# The climate of a retrograde rotating earth

Uwe Mikolajewicz [1], Florian Ziemen [1], Guido Cioni [1,2], Martin Claussen [1,3], Klaus Fraedrich [1,3], Marvin Heidkamp [1,2], Cathy Hohenegger [1], Diego Jimenez de la Cuesta [1,2], Marie-Luise Kapsch [1], Alexander Lemburg [1,2], Thorsten Mauritsen [1], Katharina Meraner [1], Niklas Röber [4], Hauke Schmidt [1], Katharina D. Six [1], Irene Stemmler [1], Talia Tamarin-Brodsky [5], Alexander Winkler [1,2], Xiuhua Zhu [3], and Bjorn Stevens [1]

[1]Max Planck Institute for Meteorology, Bundesstr. 53, 20146 Hamburg, Germany
[2]International Max Planck Research School on Earth System Modeling, Bundesstr. 53, 20146 Hamburg, Germany
[3]Universität Hamburg, Meteorologisches Insitut, Bundesstr. 55, 20146 Hamburg, Germany
[4]Deutsches Klimarechenzentrum, Bundesstr. 45a, 20146 Hamburg
[5]Department of Meteorology, University of Reading, UK

*Correspondence to:* uwe.mikolajewicz@mpimet.mpg.de

**Abstract.** To enhance understanding of Earth's climate, numerical experiments are performed contrasting a retrograde and prograde rotating Earth using the Max Planck Institute Earth System Model. The experiments show that the sense of rotation has relatively little impact on the globally and zonally averaged energy budgets, but leads to large shifts in: continental climates, patterns of precipitation, and regions of deep water formation.

Changes in the zonal asymmetries of the continental climates are expected, given ideas developed more than a hundred years ago. Unexpected was, however, the switch in the character of the Euro-African climate with that of the Americas, with a drying of the former and a greening of the latter. Also unexpected was a shift in the storm track activity from the oceans to the land in the Northern hemisphere. The different patterns of storms and changes in the direction of the trades influence fresh water transport, which may underpin the change of the role of the North Atlantic and the Pacific in terms of deep water

formation, overturning and northward oceanic heat transport. These changes greatly influence northern hemispheric climate and atmospheric heat transport by eddies in ways that appear energetically consistent with a southward shift of the zonally and annually averaged tropical rain bands. Differences between the zonally averaged energy budget and the rain band shifts leave the door open, however, for an important role for stationary eddies in determining the position of tropical rains. Changes in ocean biogeochemistry largely follow shifts in ocean circulation, but the emergence of a "super" oxygen minimum zone in

the Indian Ocean is not expected. The upwelling of phosphate enriched and nitrate depleted water provoke a dominance of cyanobacteria over bulk phytoplankton over vast areas, a phenomenon not observed in the prograde model.

What would the climate of Earth look like if it would rotate in the reversed (retrograde) direction? Which of the characteristic climate patterns in the ocean, atmosphere or land that are observed in a present-day climate are the result of the direction of Earths' rotation? Is, for example, the structure of the oceanic Meridional Overturning Circulation (MOC) a consequence of the

interplay of basin location and rotation direction? In experiments with the Max Planck Institute Earth System Model (MPI-ESM), we investigate the effects of a retrograde rotation in all aspects of the climate system.

The expected consequences of a retrograde rotation are reversals of the zonal wind and ocean circulation patterns. These changes are associated with major shifts in the temperature and precipitation patterns. For example, the temperature gradient between Europe and Eastern Siberia is reversed, and the Sahara greens, while large parts of the Americas become deserts. Interestingly, the Intertropical Convergence Zone (ITCZ) shifts southward and the modeled double ITCZ in the Pacific changes to a single ITCZ, a result of zonal asymmetries in the structure of the tropical circulation.

One of the most prominent non-trivial effects of a retrograde rotation is a collapse of the Atlantic MOC, while a strong overturning cell emerges in the Pacific. This clearly shows that the position of the MOC is not controlled by the sizes of the basins, or by mountain chains splitting the continents in unequal runoff basins, but by the location of the basins relative to the dominant wind directions. As a consequence of the changes in the ocean circulation a "super" oxygen minimum zone develops in the Indian Ocean leading to upwelling of phosphate enriched and nitrate depleted water. These conditions provoke a dominance of cyanobacteria over bulk phytoplankton over vast areas, a phenomenon not observed in the prograde model.

## 1 Introduction

When being introduced to Earth's climate, after learning about how quantities such as the latitude or elevation influence the climate of a region, school-children learn about the zonal asymmetries in patterns of weather. A common exercise in this context is to compare and contrast the climate of a city on Europe's Atlantic coast with one at a similar latitude and elevation on the Atlantic coast of North America. In the mid-latitudes, the climate of the west coast is usually more maritime and milder than on the east coast. In the subtropics, the climate of the west coast is drier and more mediterranean (winter rains) than on the east coast, where seasonality is more extreme, with more monsoonal (summer rains) patterns of precipitation.

Alexander von Humboldt (1817) was the first to document this basic asymmetry in continental climate. His zero-degree annual averaged isotherm passed through Labrador (54°N) on the western, and Lapland (68°N) on the eastern boundary of the Atlantic ocean (Munzar, 2012). These ideas were later codified by Wladimir Köppen, who – a century later – formalized his concept of climate zones (Köppen, 1923). Köppen designated north-western Europe as an Oceanic Temperate Climate (Cfb) while at the same latitude on the Eastern coast of North America, or Asia, the designation is that of subarctic (Dfc).

As Humboldt understood, the weather is only the proximate cause for this asymmetry, ultimately it is imparted by the lay and composition of the land and the sense of Earth's rotation. Winds in the midlatitudes prevail from the west – the Westerlies. In the tropics Easterlies dominate. Likewise warm western boundary currents, like the Gulf Stream in the North Altantic or the Kuroshiro in the North Pacific develop along the Eastern continental boundaries, drawing tropical waters northward before detaching from the coasts so that their seaward extensions or drifts help define the boundary between the subtropical and subpolar gyres. Their counterparts, the eastern boundary currents, flow along the opposite continental margin and advect colder subpolar waters equatorward. These cold currents are amplified by similarly flowing air currents, whose equatorward stress drives upwelling, which brings cold and nutrient rich waters from depth to the surface, giving rise to their exceptional biological productivity. A famous, and fittingly named, example is the Humboldt Current, which flows northward along the Chilean and Peruvian coast.

Contemporary scholars still debate the relative role of the winds and the currents in shaping the asymmetries in the zonal climate. Models that neglect zonal asymmetries associated with ocean currents see little change in the asymmetric continental climates, suggesting that zonal asymmetries in the climate can be explained to result from the land's influence on the atmospheric circulation (Seager et al., 2002). Yet more idealised simulations (Kaspi and Schneider, 2011) demonstrate that a simple
heat flux anomaly – similar to that which would be caused by the atmosphere flowing from a cold continent over a warmer ocean – is sufficient to set the asymmetry in the winds from which zonal asymmetries in near surface air temperatures follow. In contrast to the vast literature concerned with the meridional extent of the monsoon, or the timing of its onset, the question as to what factors influence zonal asymmetries in the locations of monsoons and deserts (Rodwell and Hoskins, 1996) has attracted much less attention.

It seems indisputable that zonal asymmetries in the distribution of the land surface combined with the sense of planetary rotation provide the ultimate reason for asymmetries in Earth's zonal climate. More disputable, and hence more interesting, is to what extent these asymmetries influence the structure of Earth's zonally averaged climate. For example, why is the zonally averaged inter tropical convergence zone (ITCZ) mostly north of the Equator? Is – as some studies argue (Wallace et al., 1989; Philander et al., 1996) – the reason primarily related to zonal asymmetries in how continental boundaries align with prevailing
wind systems? Or is its preference for the Northern Hemisphere a consequence of hemispheric asymmetries in the zonaly averaged energy budget (Chiang et al., 2003; Kang et al., 2008)? Even if the hemispheric asymmetries in the heat budget are important, this raises the question as to their origin. Is it from hemispheric asymmetries in the distribution of land masses, or from asymmetries in how the ocean transports heat, which might be more related to zonal asymmetries in the structure of ocean basins and their effect on deep-water formation.

The global ocean conveyer belt (Broecker, 1991), with meridional overturning in the North Atlantic fueling a deep western boundary current that winds through the southern ocean, around Africa and into the Indian Ocean and North Pacific where waters again upwell, is a profound example of a hemispheric asymmetry that might ultimately result from zonal asymmetries in the climate system. The drivers of the conveyor belt are still inadequately understood. Though it is clear that the salt advection feedback suggested by Stommel (1961) stabilizes the current mode of overturning, it also creates the potential for arresting the
Atlantic overturning, which thus implies multiple equilibria of the thermohaline circulation. But why is deep water not formed in the North Pacific? Two main mechanisms have been proposed (Warren, 1983), both invoking the role of zonal asymmetries. One is that the net freshwater gain in the Pacific (and the corresponding net freshwater loss in the Atlantic) leads to saltier sub-polar water in the Atlantic and rather fresh water in the sub-polar North Pacific, thus stabilizing the fresh Pacific relative to the salty Atlantic (Broecker, 1997). Another idea is that the limited northward extension of the Pacific limits the cooling in the
northernmost parts of the Pacific basin. In contrast, the Atlantic extends all the way into the Arctic allowing the surface water to be cooled down to the freezing point.

For many of the above questions, changing the direction of rotation of the Earth thus presents an easy way to obtain a completely different climate , while maintaining the familiar configuration of the continents and oceans. In the new climate, changes in wind direction lead to changes in ocean circulation, precipitation patterns and storm track locations in ways that
inform our understanding of hemispheric and longitudinal asymmetries of the present climate system, such as the position of the

tropical rain-bands, or structure of the storm tracks. Two studies (Smith et al., EGU 2008; Kamphuis et al., 2011) have explored the consequences of a retrograde rotation of the Earth with ocean circulation questions in mind. But even for these more limited in scope studies the authors come to quite different findings. Smith et al. (EGU 2008) found an inverse conveyor belt circulation with strong deep water formation in the North Pacific in a simulation with the climate model FAMOUS (Smith, 2012). Using

another model Kamphuis et al. (2011), meanwhile, found that deep water formation in the North Atlantic weakened in their simulations, and intensified intermediate water formation became evident in the North Pacific. This study did not, however, show evidence of a complete reversal in the role of sub-polar North Atlantic and Pacific for deep-water formation, leading to the conclusion that an altered net freshwater flux is not sufficient to obtain the reversal, but that the continental geometry is crucial for the pattern of the overturning circulation.

The two aforementioned studies, as noted, focus on the effect of the direction of the Earth's rotation on dynamics with a focus on the ocean circulation. We are aware of no study that has simulated the effect of changing the sense of Earth's rotation using a full Earth system model (ESM) run long-enough to reach a stationary state, and wherein most components in the Earth's oceans and biosphere (terrestrial land surface) were allowed to freely adapt to the changing conditions, and thereby define a new climate. Two hundred years after Humboldt first introduced the idea of showing climatic data on a map, and a century

after Köppen perfected this tradition, we ask: how different would his and subsequent maps have looked, had the Earth rotated in the opposite direction? In addition to shining a light on some of the aforementioned questions, these types of experiments provide important out-of-sample tests of comprehensive climate models, the results of which, if reproduced by other groups, help shape understanding of what aspects of Earth's simulated circulation systems are less sensitive to the details of how the simulation system is constructed.

**2 Model and experiments**

All simulations were performed with the ESM of the Max Planck Institute for Meteorology (MPI-ESM, version 1.2, (Mauritsen et al., submitted), as developed for use in the Coupled Modeling Intercomparison Project 6 (CMIP6, Eyring et al. (2016)). The model contains the atmospheric general circulation model ECHAM6.3.02,. This version contains some bug fixes and a different tuning, but is not structurally very different from the version used in CMIP5 (Stevens et al., 2013). The land component

JSBACH.3.10 (Reick et al., 2013) includes a dynamic vegetation model. Marine components are the ocean general circulation model MPIOM-1.6.2p3 (Jungclaus et al., 2013) and the marine biogeochemistry model HAMOCC (Ilyina et al., 2013; Paulsen et al., 2017). The coarse resolution (CR) configuration of MPI-ESM is used. It consists of ECHAM6 run with a T31 spectral truncation and with 31 vertical hybrid (sigma-pressure coordinate) levels reaching up to 10 hPa. In physical space, the spectral transform grid on which parameterized processes are solved corresponds to 96 by 48 horizontal grid cells on a Gaussian

grid. MPIOM has a curvilinear grid, with the poles located on Greenland and Antarctica. It has 120 by 101 grid cells in the horizontal and 40 levels in the vertical. A dynamic-thermodynamic sea ice model is included in MPIOM (Notz et al., 2013). The atmosphere and ocean are coupled once a day.

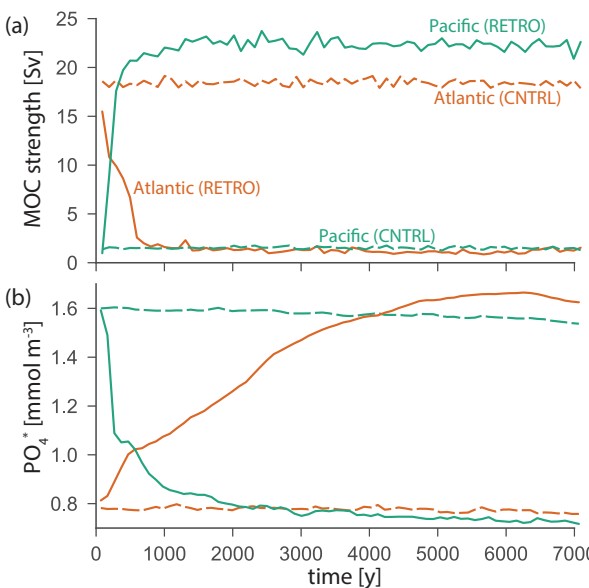

**Figure 1.** Time series (100 year means) of meridional overturning circulation for Atlantic (orange) and Pacific (turquoise) at 30°N (a); A water mass tracer (PO$_4$*) at two locations in the northern Atlantic (mean of 60°-70°W, 28°-38°N) and northern Pacific (mean of 150°-160°W, 28°-38°N), (b). RETRO is shown by the solid lines and CNTRL by the dashed lines in both panels. PO$_4$* (PO$_4$ + O$_2$/172 - 1.5) combines concentrations of phosphate and oxygen in such a way that, to first order, biological impacts are eliminated and thus can be used as a (quasi) conservative property, which in the prograde world is used to track the contributions of North Atlantic deep water (NADW) and Antarctic bottom water (AABW) to the ventilation of the deep Pacific and Indian Ocean (Rae and Broecker, 2018).

The forcing applied is the same as for the CMIP5 preindustrial control run (Giorgetta et al., 2013) and corresponds to 1850 climate conditions (fixed greenhouse gas conditions, insolation, aerosols, ozone). All simulations were started from an equilibrated climate state, resulting from a long 8000 year spin-up with some minor changes in the marine biogeochemistry module during the run. The control run CNTRL is simply a continuation of the spin-up and represents Earth's preindustrial

5 state. In experiment RETRO, representing the retrograde rotating Earth, the sign of the Coriolis parameter was changed both in the atmospheric and oceanic model components. Additionally, the direction of the sun's diurnal march was also reversed in the calculations of radiative transfer, thereby making sunrise consistent with the sense of the planetary rotations. Each simulation was integrated for 6990 model years. Whereas most physical variables were already reasonably well equilibrated after 2000 years (Fig. 1a), it took much longer for biogeochemical tracers to come close to equilibrium (Fig. 1b). Some slight drift in

10 some water column inventories does, however remain, due to the interactive sediment processes whose timescales are many millenia.

Despite the ambition to simulate the Earth system in a comprehensive manner, in a few aspects, the imprint of Earth's present day climate was prescribed. Ice sheets, greenhouse gases and aerosols were prescribed according to their present-day extent and preindustrial concentrations, respectively. The aerosol prescription means that dust deposition data used to

run the ocean biogeochemisty model reflects present day estimates. Soil colour and type were left unchanged in the land surface module JSBACH. The anthropogenic land use was prescribed according to 1850 conditions. These shortcomings in the model/experimental set-up will be dealt with in future experiments.

The experimental setup was oriented on pre-industrial (piControl) conditions, as this run served as spinup for CNTRL. Whereas the relative complete (with respect to the components) ESM was run long enough to reach equilibrium for the climate, some biogeochemical components revealed somewhat longer time scales as discussed above. Hence, for the analyses, the last 1000 years of each experiment were used. An exception is for the analysis of the atmospheric short term variability, which requires 2-hourly model output, and is based on the last 100 years of the experiments for which high-temporal resolution output was retained.

## 3   The atmosphere, its energy budget and the surface climate

At a first glance, changing the sense of planetary rotation incurs changes that Humboldt, or even Hadley (1735), might well have anticipated. The patterns of surface winds and the polarity of the temperature distribution, or the inclination of the isotherms, is reversed (Fig. 2). The change in the winds are self evident, as the trades blow from the west in RETRO and the mid-latitudes have surface easterlies. The change in the inclination of the isotherms is perhaps most evident in the distribution of the sea ice extent, this is shown by the faint blue lines in panels a and b of Fig. 2, as well as the sea-ice distributions themselves presented in Fig. 3. In RETRO, winter sea-ice extends southward enveloping Great Britain and reaching as far south as the Bay of Biscay in the Northeast Atlantic, whereas the Canadian province of Newfoundland and Labrador is free of sea-ice in winter. Similar changes are evident in both the Aleutian Basin of the North Pacific, and in the Pacific basin of the Southern Ocean. These changes are also evident in the tropical sea-surface temperatures, with the region of warmest temperatures broadening to the east in RETRO as compared to a westward broadening in CNTRL (Fig. 2).

Over the continents, the shifting isotherms mean that the Eastern continental margins warm and the western lands cool in RETRO as compared to CNTRL. In the mid-latitudes, these changes are skewed (reflecting the changing inclination of Humboldt's isotherms) so that the cooling in the western lands is poleward amplified, and the warming is strengthened toward the equator. Strong warming for instance is evident in the southeast of Brazil (Rio de Janeiro), over the southeastern states (Atlanta) of the United States of America, and over southeastern China (Guangzhou and the Pearl river delta region). The strongest cooling is over the ocean, in association with winter sea-ice, particularly over the North and Baltic Seas, although west Africa, which is very warm in CNTRL also cools substantially as does British Columbia in present day Canada. The magnitude of these changes have the effect that – from a temperature perspective – the African-European landmasses cool, and the Americas warm (Fig. 2c). Over the ocean, the cold upwelling waters evident in the present day east equatorial Pacific largely vanish, rather than shift, in RETRO. Fig. 2 shows only a hint of a cold tongue stretching eastward from East Africa in the equatorial Indian Ocean.

To a large extent, the temperature changes also reflect changes in patterns of precipitation. Circulation changes, underlying changes in the inclination of the isotherms, are evident in changes in the isohyets (lines of constant precipitation; Fig. 2). The

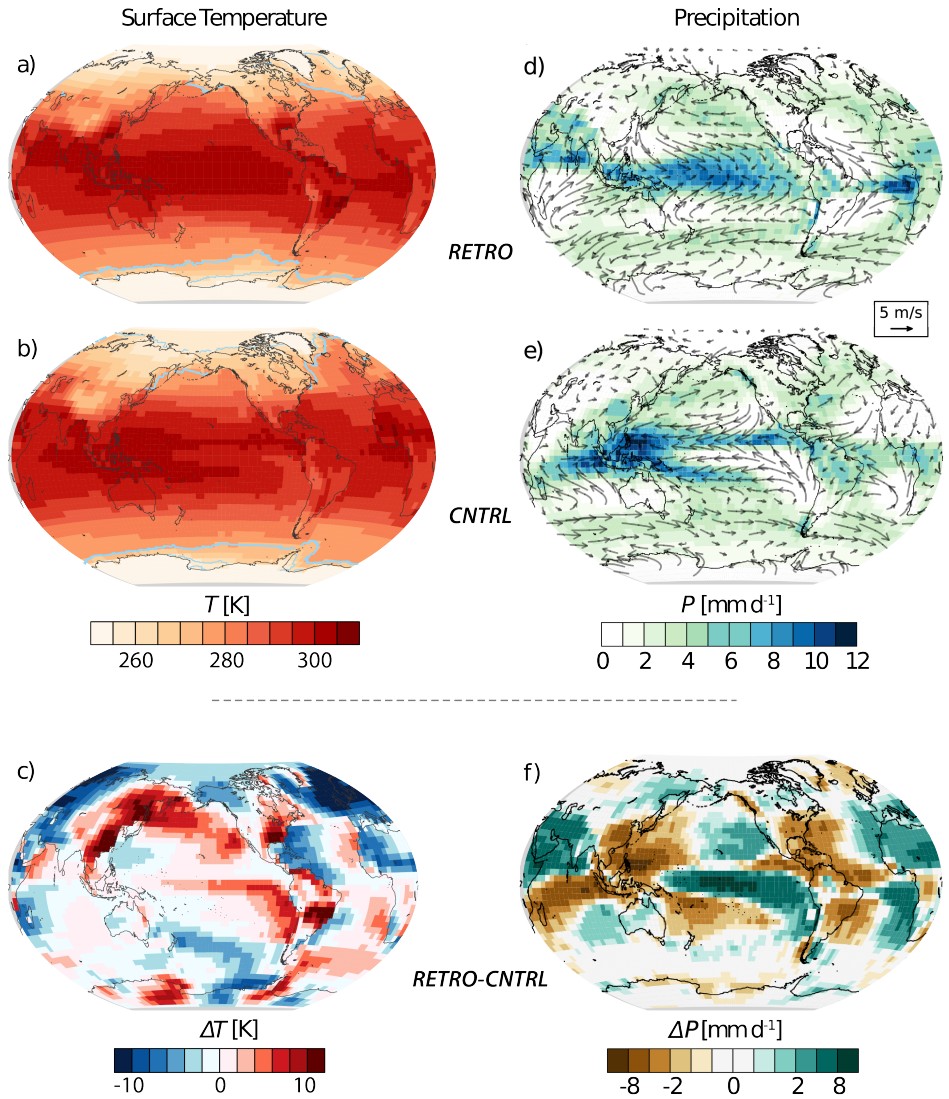

**Figure 2.** Annual averaged surface temperature in RETRO (a), CNTRL (b), and the difference between both simulations, (c). The light blue lines indicate the locations of 1 and 11 months of sea ice coverage; Annual precipitation in RETRO (d), CNTRL (e), and the difference between the two simulations (f). The curly vectors depict the annually averaged 10m wind. The seasonal cycle is shown in a supplemental video .

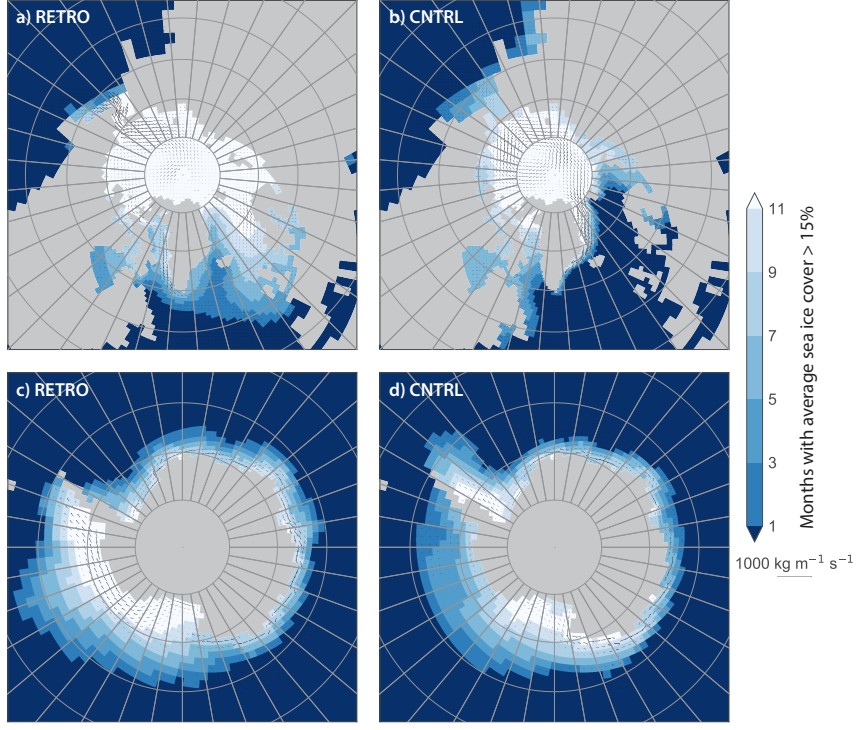

**Figure 3.** Duration of sea ice cover in RETRO (a, c) and CNTRL (b, d). The vectors show annual mean sea ice transport. For technical reasons no sea ice transport is displayed near the North Pole. The seasonal cycle of sea ice coverage is shown in a supplemental video.

simulated warming of the southeastern regions of the continents coincides with marked drying, as precipitation is displaced to areas of present-day deserts in RETRO (Fig. 2d,e). As was the case for temperature, some extra-continental scale changes are evident as again the roles of the Americas and the Afro-European land masses are exchanged. The latter becomes substantially wetter as a whole, particularly over Africa and the Mediterranean, and the former substantially drier (Fig. 2f). Precipitation in the tropics shift from the western oceans, where it is stronger north of the Equator, particularly in the Pacific, to the eastern oceans and then south of the Equator, as in the Atlantic. This results in the zonally averaged precipitation becoming slightly stronger south of the Equator, a point that we rejoin below.

RETRO also exhibits substantial changes in its monsoons and deserts, something that might have been anticipated based on earlier work by e.g. Rodwell and Hoskins (1996). Whereas in CNTRL tropical precipitation, and the velocity potential (at 200 hPa) to which it is related, is strongly focused around the western Pacific warm pool and Asian-Australian monsoon complex, in RETRO there is a greater dislocation between the land and ocean influenced precipitation. Over Asia the monsoon is displaced westward in RETRO, centering over the Arabian Peninsula. Atmospheric convection over the oceans, meanwhile, follows the warm waters to the east in RETRO, and becomes more prevalent over the central and eastern Pacific and the eoutheastern Atlantic. The area with the largest amount of annual precipitation is located near Ascension Island in the southern tropical Atlantic (8°S, 14°W), which takes on a climate more like present day Palau (8°N, 134°E). Overall the dislocation between oceanic (warmpool) and terrestrial (monsoon) precipitation in RETRO results in a more tripolar, rather than mono-polar, tropical precipitation pattern, with distinct centers of precipitation over the Middle East, extending into North Africa, over the southeast Atlantic, and into the central Pacific. These changes are reflected in the 200 hPa velocity potential (not shown), which becomes seasonally more varying and less mono-polar in RETRO as compared to CNTRL.

To assess changing extra-tropical precipitation patterns we have also tracked individual extra-tropical cyclones, defined as local minima of mean-sea-level pressure, in both RETRO and CNTRL. The track densities obtained during DJF in the northern hemisphere are presented in Fig. 4. Additional results for the tracking of cyclones during JJA in the southern hemisphere are shown in Fig. E2. As expected given the changed sense of the zonal winds, the storms track westward, rather than eastward in the mid-latitudes. This analysis (Fig. E1) shows that change in their characteristics (such as number, lifetime, growth rate or intensity) are small, although a tendency for slightly (2 hPa) more intense storms in RETRO is evident. This may be related to strengthening meridional temperature gradients, as discussed near the end of this section. The tracks of storms, however, change more substantially as they move from regions of the western-boundary currents in CNTRL to become more land centered (RETRO) in the northern hemisphere, with particularly strong storm activity extending from the Caspian Sea and Arabian Sea through the northern Mediterranean (see supplemental videos). This has large implications for the climate of North Africa and the Middle East but also acts to freshen the Atlantic at the expense of the Indo-Pacific Oceans, with implications for the meridional overturning circulation (see section 5). Further to the west, storms are most pronounced along the northern boundary of the great lakes (Fig. 4), east of the continental divide.

Changes in the storm tracks and trade winds lead to significant changes in the atmospheric energy transport, specifically in the moist-static energy transport as shown in Fig. 5. Large amounts of moist-static energy are transported from the Atlantic towards North Africa and the Middle East, where they have large implications for the regional climate. This transport persists

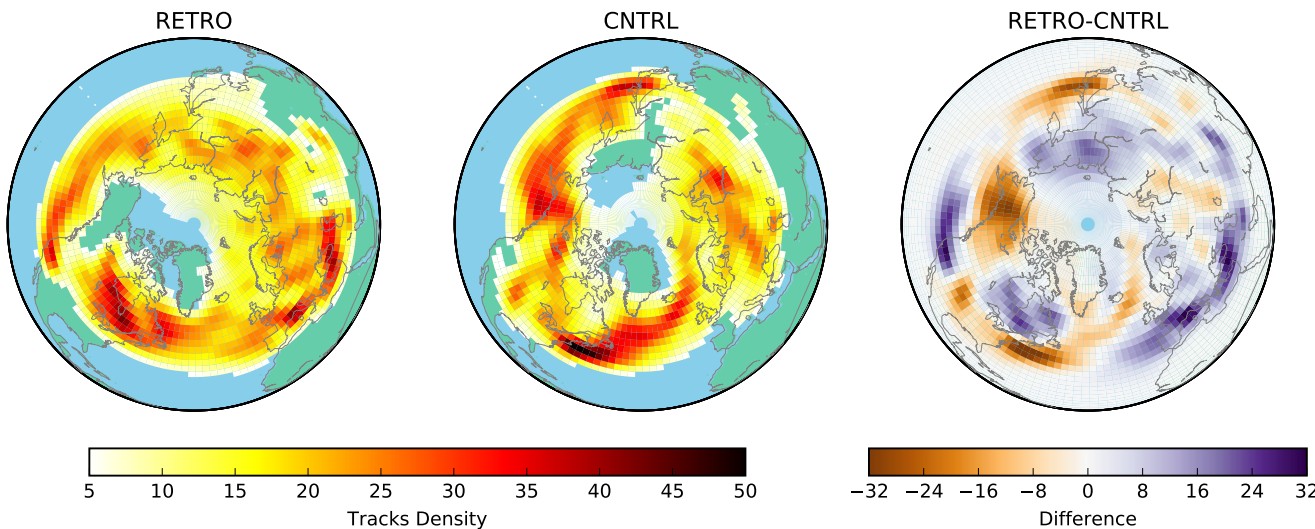

**Figure 4.** Track density (number of tracks per season for a unit area equivalent to a 5 degree spherical cup) computed for the northern hemisphere during DJF season. The tracking is performed using 2 hourly mean-sea-level pressure (MSLP) data; storms are defined as local minimum of MSLP and are tracked during their lifetime using the methodology described in Hoskins and Hodges (2002). The data are seasonally averaged over the last 100 years of each simulation. In the first two plots on the left track density for RETRO and CNTRL is shown; values below 5 are masked for the sake of visualization. Shown in the rightmost plot is the difference between RETRO and the CNTRL simulation without any masking applied. For the southern hemisphere version of this figure, see Fig. E2.

through all seasons, except winter, when the trade winds are deflected southward across Nigeria towards equatorial Africa (not shown) extending into the Indian Ocean. Further, moisture is transported from the Indian Ocean across India, Pakistan and the Middle East towards the Mediterranean Sea during all seasons except fall (Fig. 5). The enhanced moisture flux towards North Africa and the Middle East from the Atlantic and Indian Ocean provides a proximate explanation for the significant greening

in these areas (section 4). It also indicates a strong interplay between the trade winds over the Atlantic and the storm track over the Middle East.

Changing the direction of the diurnal path of the sun has only minor influence on the climate, but contributes to a southward shift of the ITCZ over the tropical Atlantic and a wetter climate over the Sahara. This is shown in appendix B, and confirms that that major circulation changes seen in RETRO versus CNTRL are ultimately dynamically driven through changes in the

10 sign of Earth's absolute angular momentum.

Despite large regional shifts in circulation patterns, the globally averaged energy budget of RETRO hardly differs from that of CNTRL, and those differences that do emerge tend to be smaller than our ability to match the energy budget as derived from observations (Stevens and Schwartz, 2012; Stephens et al., 2012). Fig. 6 is adapted from Stevens and Schwartz (2012) and shows that RETRO is slightly cooler (by about $0.14\,°C$, see Table 1, with less upwelling terrestrial radiation from the surface),

cloudier and wetter. The increase in cloudiness is evident in a slight increase in planetary albedo, with an additional $1\,\mathrm{W\,m^{-2}}$

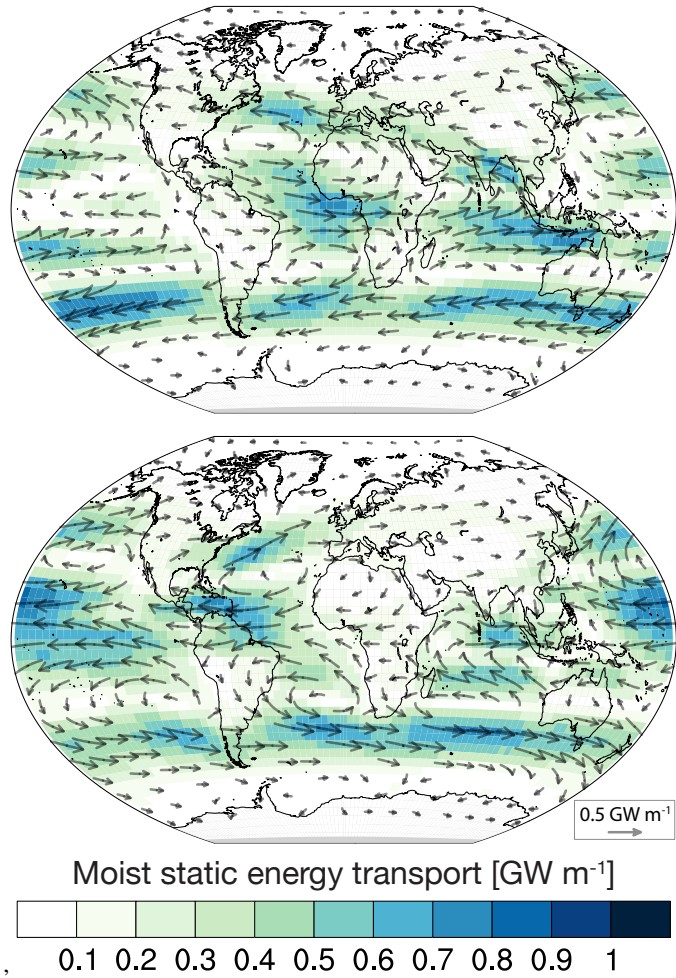

**Figure 5.** Moist static energy transport for RETRO (top) and CNTRL (bottom). To calculate the transports, we use 2 hourly data of velocities, surface pressure, temperature and specific humidity (Keith, 1995). The data are averaged over the last 100 years of the simulation.

of reflected solar radiation. Surface turbulent moisture fluxes are larger and sensible heat fluxes are smaller in RETRO. These changes are consistent with more terrestrial radiation to the surface, indicative of an atmosphere that cools and hence rains more. The surface cooling in RETRO is manifested entirely in the northern hemisphere, as the southern hemisphere actually warms by $0.28\,°\mathrm{C}$. In RETRO the difference between the average temperature of the northern and southern hemisphere is $-0.40\,°\mathrm{C}$, as compared to $0.57\,°\mathrm{C}$ in CNTRL. Concomitantly, almost all of the change in the downward longwave ($5.7\,\mathrm{W\,m^{-2}}$) is in the northern hemisphere, and almost all of this is over land.

Changes in the mean energy budget could be expected to affect the zonal distribution of precipitation. Building on the ideas developed by Kang et al. (2008, 2009) and Frierson et al. (2013), Bischoff and Schneider (2014) argue that an increase of energy release into the tropical atmosphere would shift the zero crossing of the vertically and zonally averaged moist-static

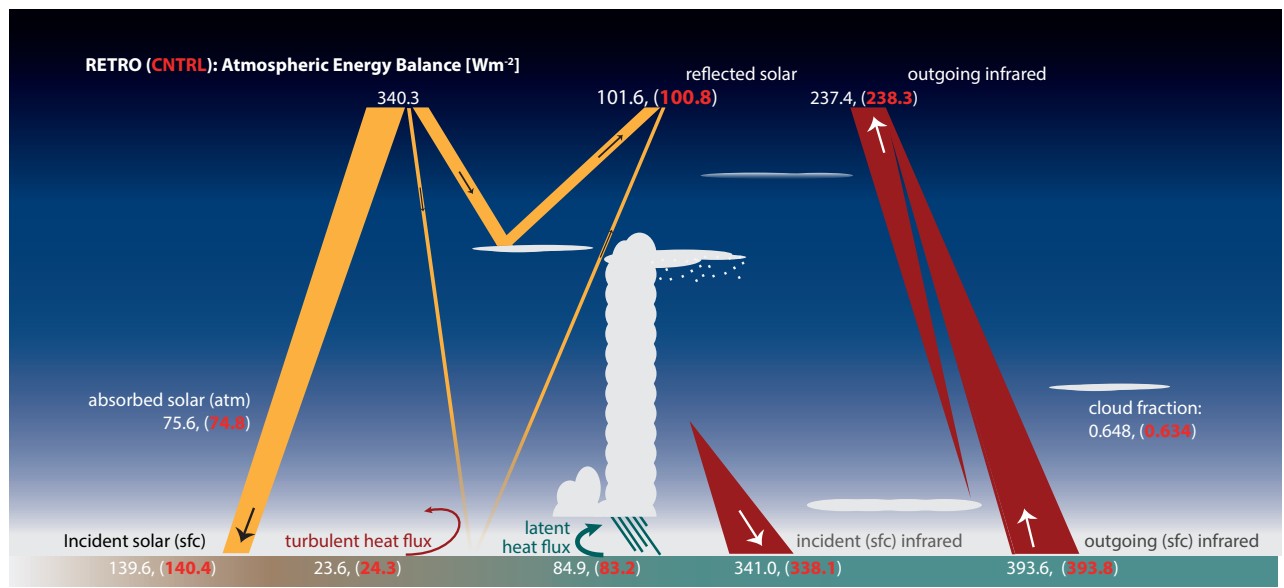

**Figure 6.** Globally averaged energy budget contributions for RETRO and CNTRL.

**Table 1.** Changes in precipitation and surface temperature. For these entries the tropics are defined as between 22.3°S and 22.3°N

|  | Global | NH | SH | Tropics |
|---|---|---|---|---|
| **Surface Temperature [K]** |  |  |  |  |
| RETRO | 287.42 | 287.21 | 287.64 | 298.98 |
| CNTRL | 287.56 | 287.76 | 287.36 | 298.53 |
| | | | | |
| **Precipitation [mm d$^{-1}$]** |  |  |  |  |
| RETRO | 3.11 | 3.18 | 3.05 | 4.09 |
| CNTRL | 3.04 | 3.04 | 3.04 | 4.05 |

energy transport (the energy flux equator) equatorward. Likewise, this way of thinking suggests that the strong increase in the RETRO northern hemisphere meridional temperature gradients implies a strengthening of the northern hemispheric energy fluxes which should also be accompanied by a southward shift of the ITCZ. Such a shift in the ITCZ is indeed pronounced in the simulations (Fig. 7a). The shift is also predicted by the change in the energy budget equator (Fig. 7b, and inset), although not

5  by the magnitude of the shift, nor does its position correlate well with the actual position of the precipitation maximum. These discrepancies might be related to changes in the zonally asymmetric circulation as earlier argued by Wallace et al. (1989) and Philander et al. (1996). To the extent the changes in precipitation are consequences of the changed hemispheric energy budget, they point to a possible role for ocean circulation, and its disproportionate impact on the northern hemispheric ice sheets and

temperature gradients, particularly over the North Atlantic. Reversing the sense of the planetary rotation does not, however, affect the distribution of land masses, so that differences between RETRO and CNTRL suggest that hemispheric asymmetries in the distributions (as opposed to the shape) of continents (i.e., the fact that Antarctica is in the southern hemisphere) is at least not sufficient to explain why the ITCZ is mostly north of the equator.

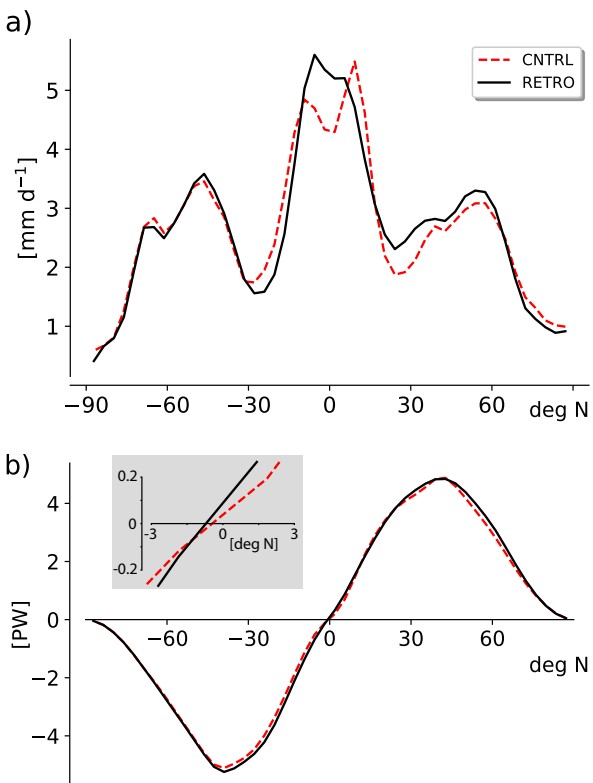

**Figure 7.** Zonally and annually averaged (a) precipitation; and (b) atmospheric energy flux for RETRO (black, solid) and CNTRL (red, dashed). The inset in panel (b) shows the atmospheric energy flux (computed as the integral of the vertical heat flux convergence as a function of latitude) in the vicinity of the zero crossing near the equator.

5   Many of the points discussed above are amplified by an inspection of the latitude-height description of the zonally averaged circulation. Fig. 8a shows the differences between RETRO and CNTRL of annually and zonally averaged temperatures. Fig. 8b presents the corresponding annually and zonally averaged zonal winds. The surface temperature anomaly pattern shows the tropical warming in RETRO and the strong northern hemispheric cooling. The subtropical jets in RETRO are shifted north-wards, i.e., equatorwards in the SH and polewards in the NH, and slightly increased in magnitude. This is consistent (through

10  the thermal wind balance) with changes in tropospheric temperatures whereby the high-latitude cooling in RETRO is much more pronounced in the northern hemisphere. This pattern of tropical warming and high-latitude cooling implies greater baro-

clinicity at mid-latitudes, but is accompanied by relatively little change in storm activity (see above), suggesting that enhanced northern hemispheric energy fluxes are mostly attributable to the changing enthalpy gradients.

As typical for climate model simulations of global warming, the tropical warming is stronger in the free troposphere than at the surface and reaches about $1.7\,\mathrm{K}$ around $300\,\mathrm{hPa}$. Also typical for a globally warmer tropical troposphere is the increase of the cold-point tropical tropopause, i.e., the coldest point in the tropical tropopause region, as one might also expect if the tropopause temperature is radiatively controlled following the ideas of Zelinka and Hartmann (2010), which in RETRO is about $0.6\,\mathrm{K}$ warmer than in CNTRL.

A warmer cold-point tropopause and the resulting increased water vapour entry into the stratosphere is the likely cause of an about $13\,\%$ larger specific humidity in the lower to middle stratosphere in RETRO (not shown). More stratospheric water vapour is a plausible reason for the, in general, lower stratospheric temperatures.

Changing the planetary rotation should, of course, to a first order cause a reversal of zonal winds. Fig. 8b shows the sum of annually and zonally averaged zonal winds of RETRO and CNTRL, which indicates 2nd-order effects, i.e., deviations from a simple change in sign. Changes in the stratospheric circulation should be interpreted with caution due to the model top in the middle stratosphere at $10\,\mathrm{hPa}$. However, the strongest stratospheric cooling, which occurs in the polar northern hemisphere, is dominated by a cooling in boreal winter, which further contributes to northern hemispheric baroclinicity, and is related to a stronger polar vortex in RETRO. The average eddy heat flux entering the stratosphere in boreal winter between $40°\mathrm{N}$ to $80°\mathrm{N}$ is weaker in RETRO by about $15\,\%$ (not shown), thereby helping to explain the temperature changes required to balance this stronger polar vortex. The change in tropospheric flow patterns are thus connected to the stratosphere. The latitude dependence of the temperature signal in the southern stratosphere, although consistent with changes in vertical motion, is less straightforward to interpret.

In two sensitivity experiments (abrupt quadrupling of atmospheric $CO_2$, CNTRLx4 and RETROx4) the climate sensitivity was investigated. Details are given in appendix C. In general, RETROx4 showed a higher climate sensitivity (approximately 10% to 15%). In spite of the large differences in tropical climate, the strength of the tropical cloud effect seems to be rather similar between the prograde and the retrograde simulations. A striking difference appears in the long-term equilibration (years 1000-2500), where CNTRLx4 shows a change towards much faster equilibration at a lower level which is absent in RETROx4. This difference is connected to differences in the heat uptake of the deep ocean.

## 4 Land surface and biosphere

Köppen, in his book *Die Klimate der Erde* (1923), outlined his ideas regarding how continents impart zonal asymmetries in climate using his classification system applied to an idealized continent. For the purposes of the present discussion, Köppen's idealized continent has been redrafted, and is presented in Fig. 9. The expectation inherent in this idealization is that a retrograde rotating Earth would experience mirror symmetry about the North-South axis in its climate zones.

As might have been expected given the discussion of the previous section, predictions based on Köppen's idealized continent, both for the distribution of climate zones in the present day climate, and for the expected mirror symmetry for a retrograde

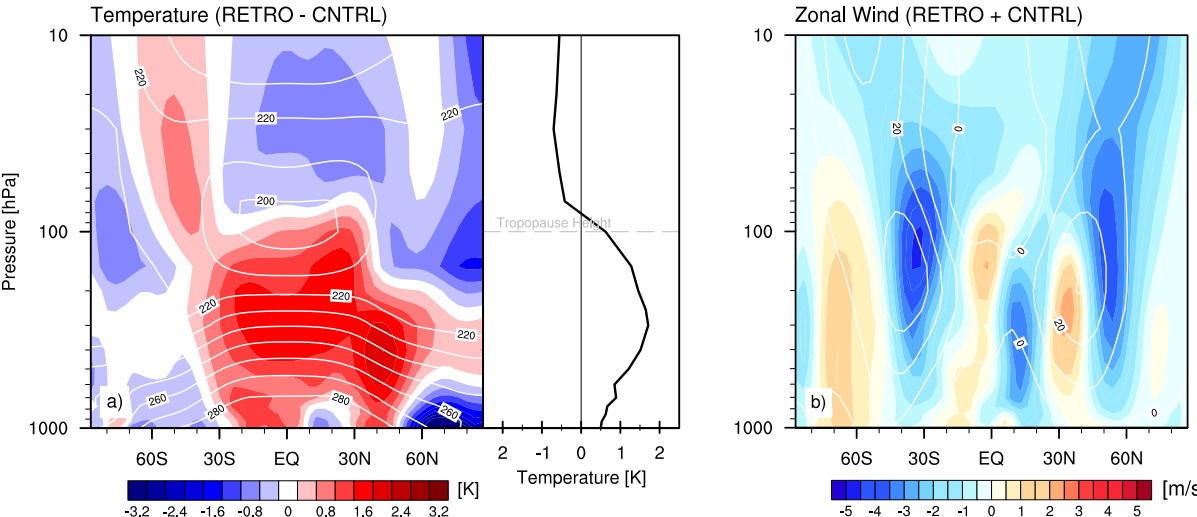

**Figure 8.** (a) Annually and zonally averaged temperature difference between RETRO and CNTRL (left) and its tropical (20°S to 20°N) average (right). The tropopause level indicated in the right panel refers to the cold-point tropopause of the control case. (b) Annually and zonally averaged zonal wind sum of RETRO and CNTRL. White contour lines show the isotherms and isotachs of the control simulation with a contour interval of (a) 10 K and (b) 10 m/s.

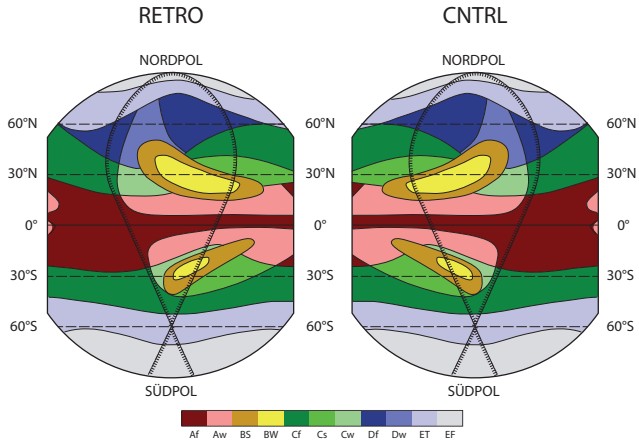

**Figure 9.** Classification of climate zones for an idealized continent, the Klimarübe, following Köppen (1923). First letters in the legend are main climate types: A: equatorial, B: arid, C: warm temperate, D: snow, E: polar. Second letters are precipitation regimes: W: desert, S: steppe, f: fully humid, s: summer dry, w: winter dry, T: tundra, F: ice cap.

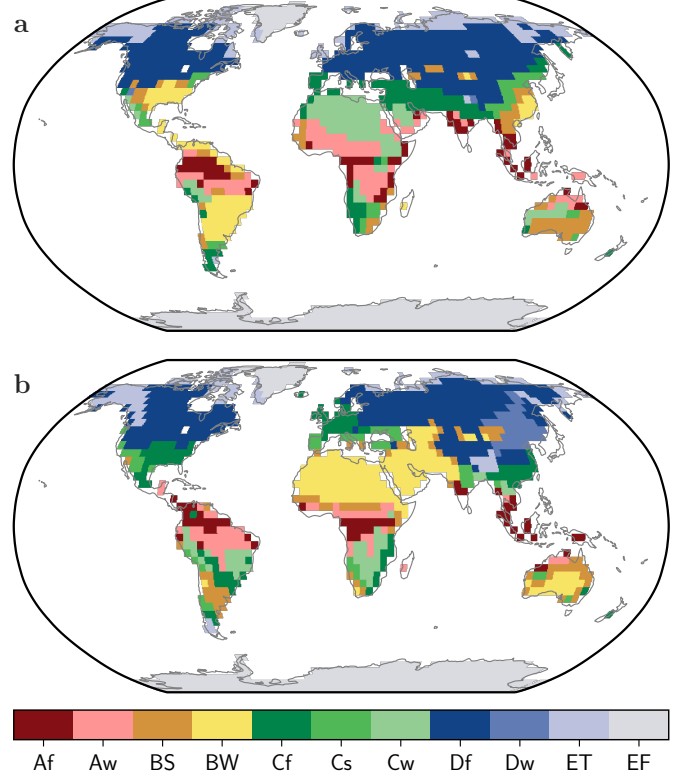

**Figure 10.** The eleven main climate zones after the Koeppen-Geiger classification (colors) for a) RETRO and b) CNTRL. First letters in the legend are main climate types: A: equatorial, B: arid, C: warm temperate, D: snow, E: polar. Second letters are precipitation regimes: W: desert, S: steppe, f: fully humid, s: summer dry, w: winter dry, T: tundra, F: ice cap

rotating Earth are well supported by CNTRL and RETRO. This is shown, using the same indication of major climate zones, in Fig. 10. CNTRL reproduces both the North-South asymmetry associated with more northern hemispheric land masses as reflected by the emergence of cold winter climates (Dw and Df) in the northern hemisphere, and the east-west asymmetry of a variety of features. For instance the shift from subtropical deserts (BW) in the continental southwest toward moist temperate

5   climates in the continental southwest at subtropical latitudes is well evident in the contrast between West Africa and Southeast Asia. To a first approximation in a retrograde rotating Earth these features do appear with mirror symmetry (Fig. 10b).

There are however some differences that would not have been predicted from just mirroring the idealized continent. Most prominent is the shift in deserts from the Eurasia-African continental mass to the Americas, with a greater center of mass over the subtropical southern hemispheric continents. This change is consistent with the inferences of the last section, whereby in

10   RETRO many of the climate features associated with present day Europe/Africa and North/South America are exchanged.

The complete replacement of the wide desert belt from West Africa to the Middle East by forests and humid grasslands is more quantitatively measured by changes to the leaf-area index (LAI, Figure 11). Changes in LAI also illustrate the degree to

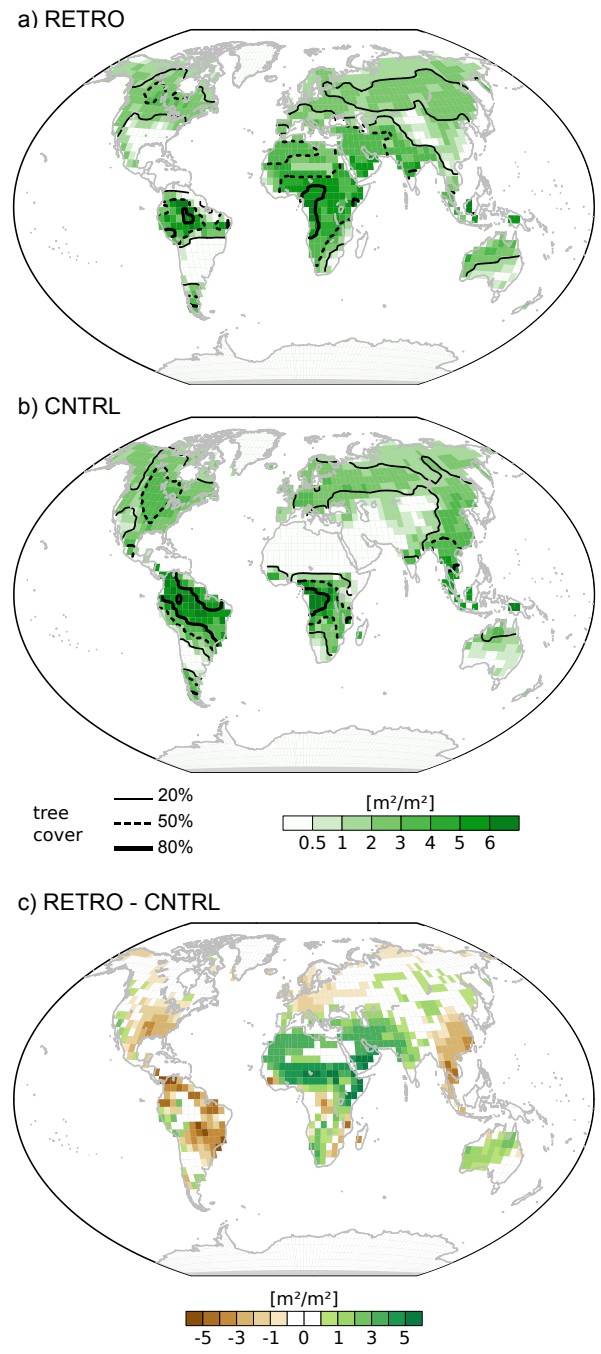

**Figure 11.** LAI (shaded) and tree cover (contours) for a) RETRO and b) CNTRL. c) shows the differences in LAI (see also supplemental video).

which the retreat of desert climates in Africa and Eurasia is accompanied by an extensive formation of dry climates in South and North America. Southern Brazil and Argentina become the Earth's biggest deserts and the southern States of the United States of America see a dramatic climate shift from a fully humid climate towards a complete aridification. In general, many dry regions are simulated in RETRO, but extreme deserts – like the present-day Sahara – are less widespread. Changes that are, as discussed in section 3, consistent with circulation and precipitation changes over these same regions.

The global area covered by permanent deserts is reduced by about $25\,\%$ from $42 \times 10^6\,\mathrm{km}^2$ to $31 \times 10^6\,\mathrm{km}^2$ (see Table 2). Woody and herbaceous vegetation fill the new vegetated areas in about equal measure. The greening is concentrated over northern hemispheric land masses (desert areas shrink by nearly 40%) and mostly attributed to the aforementioned vanishing of the wide desert belt from West Africa to the Arabian peninsula. Over southern hemispheric land masses, deserts and grasslands spread slightly, whereas the tree-covered area is reduced by about $20\,\%$. Tropical vegetation remains largely unaffected in Africa and Asia. In South America, the Amazon rain forest shrinks substantially, though.

**Table 2.** Global area of main vegetation groups averaged over the last 1000 years of experiment.

| Area covered [Mio. km$^2$] | RETRO | CNTRL | diff |
|---|---|---|---|
| **Permanent deserts** | | | |
| global | 31.2 | 41.8 | -10.6 |
| NH | 21.0 | 34.1 | -13.1 |
| SH | 10.2 | 7.7 | +2.5 |
| | | | |
| **Woody vegetation** | | | |
| global | 46.3 | 41.3 | +5.0 |
| NH | 33.4 | 25.0 | +8.4 |
| SH | 12.9 | 16.3 | -3.4 |
| | | | |
| **Herbaceous vegetation** | | | |
| global | 50.0 | 44.4 | +5.6 |
| NH | 41.1 | 36.4 | +4.7 |
| SH | 9.0 | 8.0 | +1.0 |

The greener climate in RETRO affects the global carbon storage on land, which increases by $86\,\mathrm{PgC}$ compared to the CNTRL simulation. The globally integrated storage increase is composed of an increase by $215\,\mathrm{PgC}$ in the northern hemisphere and a decrease of $130\,\mathrm{PgC}$ in the southern hemisphere.

Based on previous experience (e.g., Claussen, 2009; Bathiany et al., 2010) we suppose that interactions between the biosphere and the atmosphere amplify or dampen the near-surface climate changes. The reduced tree cover in the much colder and

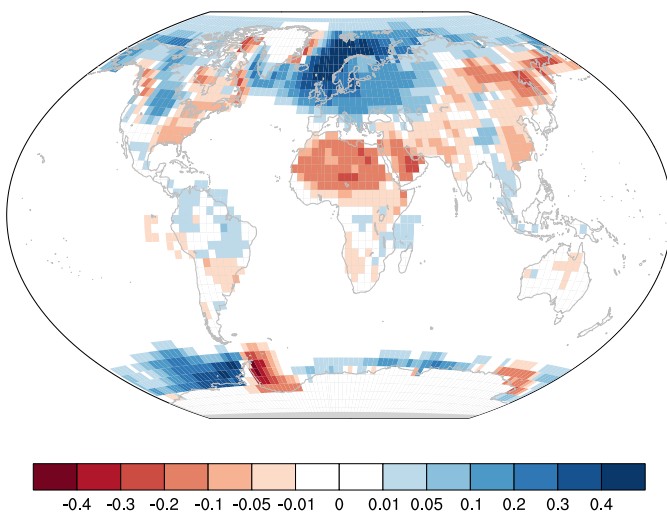

**Figure 12.** Differences in surface albedo (RETRO-CNTRL)

snowier Europe, for example, coupled with the greening of North Africa and the Middle East likely induces a large-scale and large-amplitude change to the surface albedo (Fig. 12). These changes are expected to modify the local climate directly, for instance through enhanced near surface cooling via the effect of reduced snow masking, or indirectly by inducing circulation changes. The depicted changes in surface albedo (Fig. 12) also hint at a limitation of our study. We have prescribed present

soil albedo and land use as of 1850 CE. Hence, we expect that changes of soil albedo caused by an increase in soil carbon due to plants growing in the retrograde green Sahara (see Vamborg et al. 2011) could further decrease the surface albedo in that region or, vice versa, enhance the surface albedo in the retrograde deserts of the Americas. Likewise, the atmospheric aerosol remained unchanged in the simulations, with an interactive aerosol, precipitation changes over RETRO in Africa and, hence Sahara dust emissions would likely decrease, whether this would be compensated by substantial increases in emissions from

other sources remains an open question.

Another prescribed present day feature are the ice sheets. In the RETRO and CNTRL simulations, no permanent snow cover was simulated outside of the prescribed glaciated areas (not shown). Additionally, the simulated summer temperatures over the ice sheets (not shown) do not indicate a strong inconsistency between the prescribed ice sheets and the simulated climate both for RETRO and CNTRL. Therefore, we did not find a major inconsistency between the prescribed present-day ice sheet

configuration and the simulated climate.

## 5   Ocean physics

As expected, the change of the sign of the Coriolis parameter has a marked impact on the ocean circulation (Fig. 14, see supplemental video). Directly associated with this shift is a change in the sign of the zonal ocean velocity components. Subpolar and subtropical gyres shift their longitudinal positions and what used to be a western boundary current in CNTRL becomes an

eastern boundary current in RETRO. These shifts cause significant changes in the SST patterns in the subtropical gyres. The strong poleward transport of warm water in the eastern boundary currents leads to a local warming and the missing poleward flow of warm waters on the western side of the basins leads to a local cooling (Fig. 2a and c).

The Antarctic circumpolar current (ACC) reverses its direction and moves from east to west in RETRO. The Indonesian throughflow changes it sign and transports 21 Sv from the Indian Ocean to the Pacific. Besides, many of the gyres show substantial changes in strength. In RETRO, both subtropical and subpolar gyres in the North Atlantic become weaker than in CNTRL. The main cause for this is the lack of the thermohaline driven overturning component in the Atlantic (see below), reducing the North Atlantic gyres to their wind-driven part. Further, the straight western coastline of the Americas enables the development of closed gyres in the Pacific in RETRO. In contrast, the gyres in CNTRL are affected by the openings along the highly structured Asian/Australian coastline. Especially the South Pacific subtropical gyre in RETRO becomes with more than 130 Sv approximately 2.5 times stronger than in CNTRL.

In CNTRL, Antarctic bottom water is formed in the center of the Weddell gyre. In RETRO, deep water formation in the southern hemisphere (as indicated by the maximum mixed layer depth; Fig. 14) takes place in the Atlantic sector of the Southern Ocean, close to the coast of Antarctica. On the northern hemisphere, the changes are more dramatic. In CNTRL, the model simulates an Atlantic meridional overturning circulation (MOC) of 19 Sv. North Atlantic deep water is formed in the Nordic Seas and intermittently in the Labrador Sea. In RETRO, only intermittent convection down to 2000 m depth occurs in the North Atlantic (Fig. 14). The appearance of deep convection has a multidecadal time scale with a peak around 80 years. In the long-term mean, no clear North Atlantic deep water cell emerges in RETRO. Instead, the northeast Pacific exhibits strong formation of deep water (down to 3000 m). This leads to a strong Pacific MOC of 23 Sv (Fig. 13). The core of the North Pacific deep water is slightly warmer and shallower than its North Atlantic equivalent in CNTRL. This indicates, that the Atlantic and Pacific change their roles in CNTRL and RETRO.

Whereas the global poleward oceanic heat transport on the northern hemisphere is almost unchanged (1.56 PW in CNTRL vs. 1.52 PW in RETRO at 30°N , Fig. 15 a), the distribution between the basins changes strongly. In CNTRL, more than half of the heat is transported in the Atlantic. In RETRO, the Pacific transports almost 80% of the heat. The transport in the Pacific is larger, as the gyre-transport in the Pacific is stronger than in the Atlantic, which can be explained by the larger spatial extent of the Basin. Thus, also in terms of the ocean heat transport, Atlantic and Pacific have changed their roles. Due to the strong upwelling of colder subsurface water, the tropical Indian Ocean (north of 30°S) becomes a strong heat sink for the atmosphere in RETRO (0.84 PW). This is in contrast to CNTRL, where the tropical Indian Ocean has a rather small net heat uptake (0.1 PW). In RETRO, the tropical Indian Ocean has de facto taken the role of the tropical East Pacific in CNTRL. This is also true for tropical SST variability. The leading mode in RETRO has its center of action in the Indian ocean, in contrast to the El Niño mode in CNTRL. More details can be found in appendix A. For the export of heat to the tropical Pacific the reversed Indonesian throughflow plays a key role. In RETRO, the equivalent of the cold Humboldt Current flows northward in the western Indian Ocean along the coast of Mozambique.

The changes in ocean heat transport are also reflected in SST and sea ice changes. In general, the North Atlantic is colder in RETRO, while the North Pacific is warmer (Fig. 2). The effect of the reduced ocean heat transport in the North Atlantic

is amplified by the reduced wintertime mixed layer depth, which reduces the effective heat capacity of the surface ocean and, thus, leads to colder winter temperatures and extended sea ice cover (Fig. 3).

In RETRO the Atlantic north of 30°N is a strong sink for atmospheric moisture (0.70 Sv), whereas the net flux into the North Pacific is much smaller (0.10 Sv), whereas in CNTRL these values are almost equal (Fig. 15b). The net moisture loss of the Pacific north of 30°S is 0.30 Sv, in the Atlantic only 0.07 Sv. In CNTRL the role of the two oceans is reversed. In RETRO, the Atlantic gains moisture from the Indian ocean by atmospheric moisture transport across the Middle East, which is strongest in spring and summer and loses moisture by the eastward transport across equatorial Africa. The transport across Central America is very small, but directed towards the Atlantic, whereas in CNTRL a strong transport is directed towards the Pacific (Fig. 5). Consequently, the North Pacific surface salinity in RETRO is higher than in CNTRL; in the North Atlantic surface salinity is reduced (see supplemental video). This explains the shift of the deepwater formation from the North Atlantic to the North Pacific described above.

Due to strong precipitation in RETRO, the Mediterranean and Red Sea no longer are marginal seas with net evaporation (as in CNTRL). This has dramatic consequences for the overturning circulation in these basins. Whereas both basins today (and in CNTRL) are characterized by deep convection and a deep outflow of salty water, the circulation in RETRO is completely reversed. Both basins are well stratified and the main source of deep water is the inflow. The outflow at the surface is rather fresh, typical for estuarine circulations.

The lack of deep water formation in the North Atlantic and the Arctic and the changes in density lead to an increase in sea level in the Atlantic and the Arctic by typically 0.5 m, whereas the sea level in the Pacific and the Indian Ocean decrease (not shown). This reverses the sea level gradient across Bering Strait and leads to a southward flow of fresh Arctic surface waters into the North Pacific.

The direction of the simulated changes is similar to the outcome of experiments by Smith et al. (EGU 2008) and Kamphuis et al. (2011), but in general the circulation changes found in our model are more similar to the results of Smith et al. (EGU 2008) than to the results of Kamphuis et al. (2011). Both studies showed a weakening of the Atlantic MOC together with a surface freshening in the North Atlantic and saltier surface conditions in the North Pacific. Whereas Smith et al. (EGU 2008) also showed a strong MOC in the Pacific, Kamphuis et al. (2011) obtain a state with a relative weak MOC both in Atlantic and Pacific.

## 6 Ocean biogeochemistry

Ocean circulation is the key driver of spatial patterns of marine biogeochemical tracers. This implies large scale patterns of these tracers follow the reversal of the ocean circulation (described in section 5) and biogeochemical water mass trackers, such as $PO_4^*$ mirror the transient behavior of the MOC reversal though with longer equilibration time scales (Fig 1b). Planetary-scale features such as zonal and global means are largely unaffected by the direction of Earth's rotation (Tab. 3). The changes in the Coriolis effect and wind patterns lead to a shift of eastern boundary upwelling systems to the western sides of ocean basins in RETRO (Fig. 16). In these upwelling systems, in general, primary production and, thus, carbon storage in particular

organic matter (POM) is driven by a continuous supply of nutrients from greater depth into the sun-lit surface layers (euphotic zone). Gravitational settling and subsequent remineralization of POM, a process being referred to as biological pump, lead to pronounced vertical gradients in all biogeochemical tracers. Exported organic material is remineralized under the consumption of oxygen. For this reason, hypo- or suboxid conditions are often found at mid-depths in regions with high biological production

5   such as upwelling systems. The global volume of these oxygen minimum zones (OMZs) is nearly identical in RETRO and CNTRL (Tab. 3, see also supplemental video). Despite that, in CNTRL three major OMZs are found on the eastern side of each ocean basin while in RETRO there is one sizable OMZ in the Indian Ocean. Strong upwelling, which is also reflected in strong heat uptake, fuels biological production rates at levels not found in CNTRL. In combination with the circulation and the basin geometry of the Indian Ocean this leads to very low ventilation and nutrient accumulation in the northern part of the

10   basin (Fig. 17) and results in the development of this extended OMZ. One prominent characteristic of OMZs is that in the low oxygen environment denitrifying bacteria are able to access food energy by degradation of organic matter using nitrate ($NO_3$). Denitrification is limited to very low oxygen conditions (in the model $O_2 \leq 0.5\,\mathrm{mmol\,m^{-3}}$) and is the only remineralization process that selectively removes bioavailable nitrogen. All other degradation processes, i.e. aerobic remineralization and sulfate

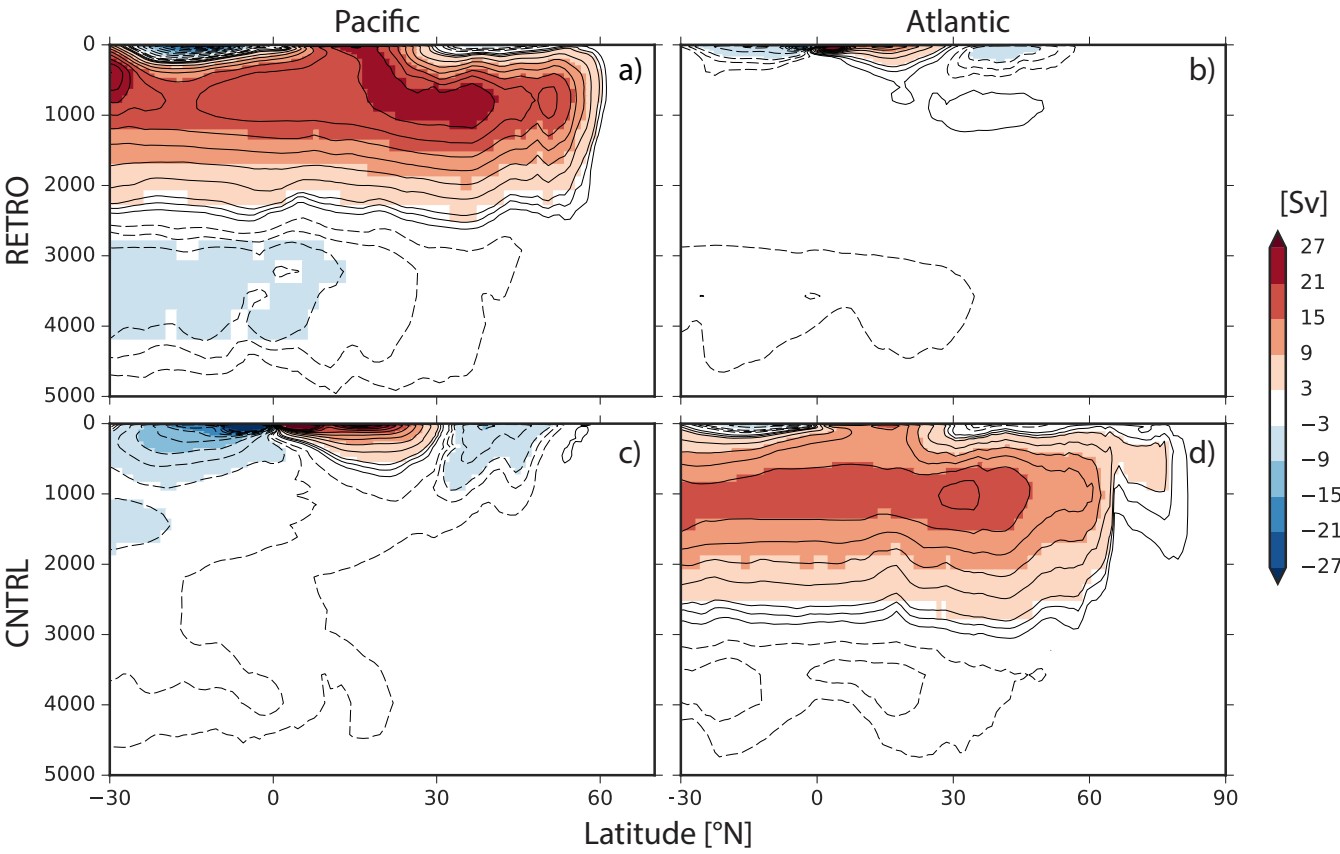

**Figure 13.** MOC stream functions. Outlines at $\pm 1, 2$ and multiples of 3 Sv. The Indian Ocean MOC stream functions are shown in Fig. 17.

reduction, convert organic material into phosphate, iron, and nitrate. In RETRO global denitrification is about 50 % higher than in CNTRL and takes place predominately in the northern Indian Ocean (Tab. 3, see also supplemental video). As a

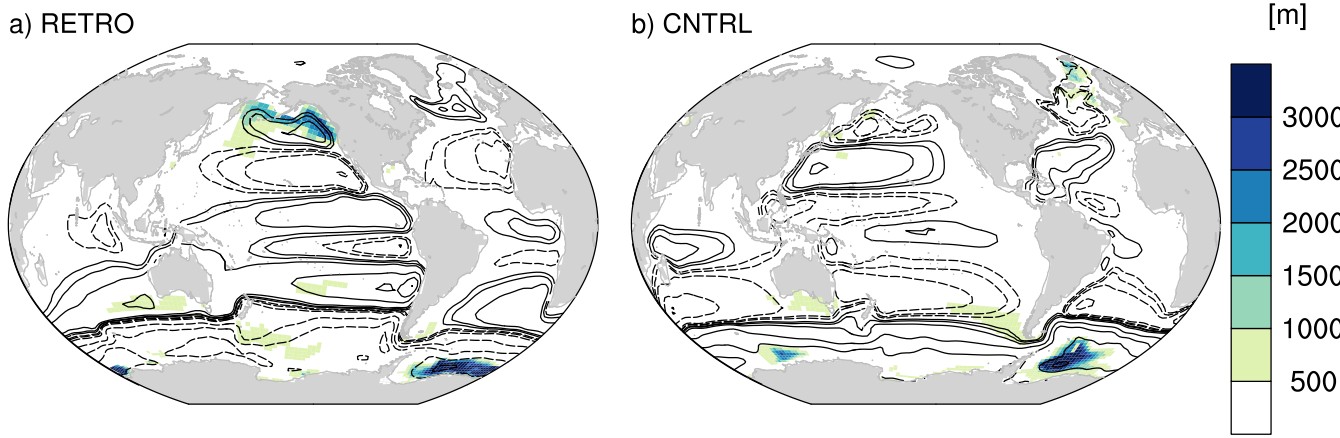

**Figure 14.** Maximum mixed layer depths and annual mean barotropic stream functions. Outlines at $\pm 5, 10, 20, 60, 100, \ldots$ Sv. a) RETRO, b) CNTRL

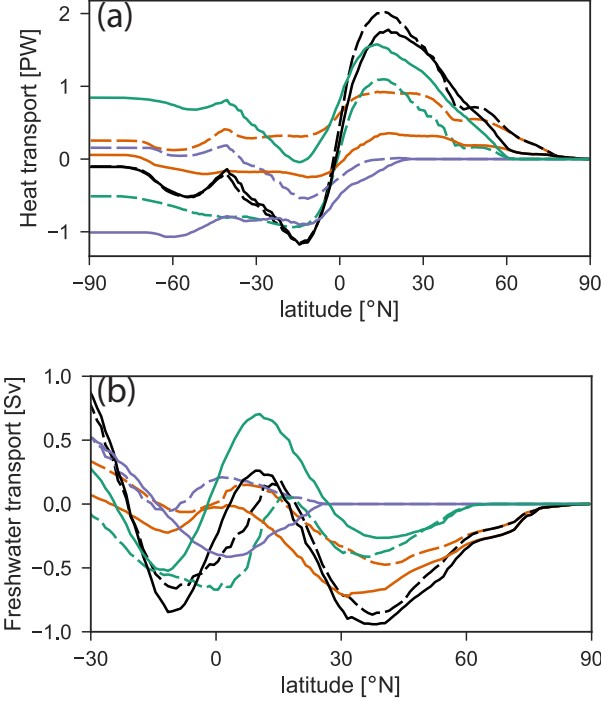

**Figure 15.** Northward implied ocean transports of (a) heat and (b) freshwater. Black: global, orange: Atlantic, turquoise: Pacific, violet: Indian Ocean. Solid: RETRO, dashed: CNTRL

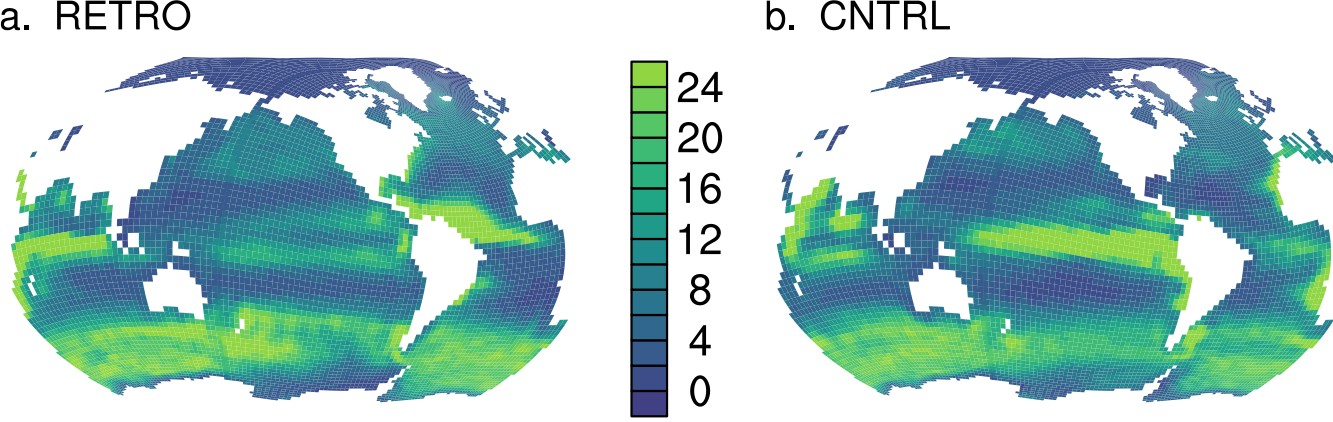

**Figure 16.** Net primary production $[\mathrm{molC\,m^{-2}\,yr^{-1}}]$ a. RETRO b. CNTRL

consequence, upwelling of nitrate depleted and phosphate rich water ($\mathrm{N^* = NO_3 - 16 \cdot PO_4 < 0}$, Fig. 18) results in a shift of the pythoplankton species composition (Fig. 17) with a dominance of cyanobacteria in RETRO. Most pythoplankton species need both nitrate and phosphate for their growth. Only cyanobacteria are able to grow on dinitrogen ($\mathrm{N_2}$), as long as sufficient phosphate and iron are available (e. g. Sohm et al., 2011). In the prograde world, only few regions exist, where the surface water
is nitrate depleted and phosphate rich. Thus, cyanobacteria are nearly everywhere outcompeted by other phytoplankton species (bulk phytoplankton). In contrast, in RETRO they become the dominant primary producer in the northern Indian Ocean. The change of the Earth's rotation direction and the subsequent development of an extended oxygen minimum zone in RETRO provoke plankton species compositions over large areas which have not been observed in the prograde world.

It was hypothesized that a warming climate and the consequential deoxygenation of the ocean (e.g. Breitbarth et al., 2007;
Hutchins and Fu, 2017) trigger such an ecosystem shift with nitrogen-fixing cynaobacteria as a potential winner. Thus, the response nicely supports the possibility of these extreme changes in the ecosystem. It also demonstrates the model's ability to adapt to an unconventional forcing and to simulate phenomena which are a result of complex interaction of abiotic and biotic processes.

The model setup, of course, includes simplifications that might affect characteristics of the response. For example, in lack
of a dynamical aerosol model, we use dust deposition maps derived from a prograde model as a source of dissolved iron. Thereby, large ocean inputs occur downstream major deserts, such as from the Sahara into the Atlantic. A displacement of deserts as simulated by the land model in RETRO (see Sec. 4) would also imply a shift of dust deposition maximums. This might affect the relative importance of P- and iron-limitation of cyanobacteria (as described in Sohm et al., 2011). Hence, the availability of dissolved iron in RETRO is controlled by ocean circulation patterns and upstream consumption by plankton in
regions where local aeolian supply is low owing to the fixed (prograde) dust deposition field. On the contrary, some regions like the Northern Indian Ocean receive potentially a higher iron supply by this dust field. However, the dominance of cyanobacteria in the Northern Indian Ocean in RETRO is primarily a result of the interplay between the high equatorial primary production

**Table 3.** Global mean net primary production rate, biogenic material export flux at 100 m, volume of OMZ, global mean denitrification and ratio of denitrification to organic matter export flux a on percentage basis $R_{den}$

| | Net primary production [GtC yr$^{-1}$] | | |
|---|---|---|---|
| | bulk phyto | cyano | total |
| CNTRL | 49.97 | 3.09 | 53.05 |
| RETRO | 48.02 | 3.73 | 51.75 |

| | Export flux | | |
|---|---|---|---|
| | org. matter [GtC yr$^{-1}$] | opal shells [kmol Si s$^{-1}$] | calcite shells [GtC yr$^{-1}$] |
| CNTRL | 7.58 | 3464.0 | 0.54 |
| RETRO | 7.78 | 3422.7 | 0.49 |

| | Volume of OMZ ($10^{16}$ m$^3$) | Denitrification (Tg N yr$^{-1}$) | $R_{den}$ (%) |
|---|---|---|---|
| CNTRL | 3.90 | 109 | 1.07 |
| RETRO | 3.83 | 154 | 1.46 |

**Figure 17.** Zonal means in the Indian Ocean for RETRO (left) and CNTRL (right) : net primary production (NPP, upper row, black, [molC m$^{-2}$ yr$^{-1}$]) with contributions of bulk phytoplankton (green) and cyanobacteria (red); phosphate concentration (lower row, [mmol m$^{-3}$]). Contours of meridional overturning circulation (MOC) are overlaid at levels of $\pm$ 1, 2, and multiples of 3 Sv.

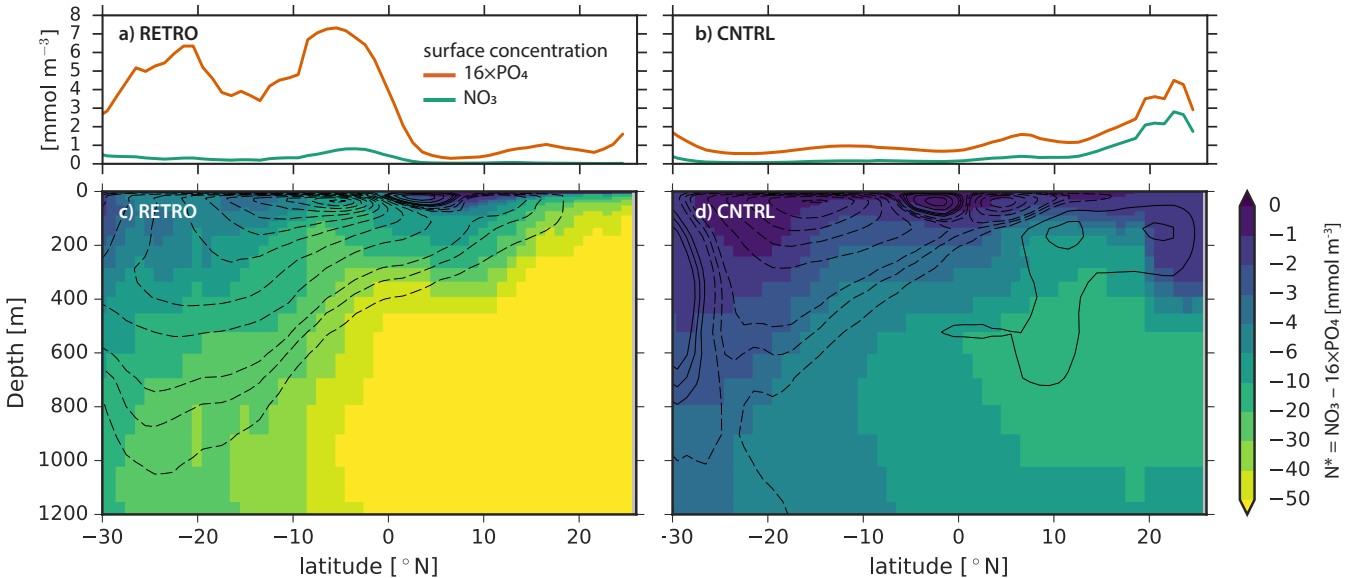

**Figure 18.** Zonal means in the Indian Ocean for RETRO (left) and CNTRL (right): nitrate (green) and phosphate ($\cdot16$, red) concentrations (upper row) $[\mathrm{mmol\,m^{-3}}]$, and $\mathrm{N^* = NO_3 - 16 \cdot PO_4}$ (lower row) in the Indian ocean. Contours of meridional overturning circulation (MOC) are overlaid at levels of $\pm 1$, 2, and multiples of 3 Sv.

creating water masses depleted in nitrate and enriched in phosphate and iron and the ocean circulation with a northward transport and subsequent upwelling of theses water masses (Fig. D1). Independent of aeolian iron supply the lack of nitrate (Fig. 18) inhibits growth of bulk phytoplankton fostering the local dominance of cyanobacteria in the Northern Indian Ocean.

Furthermore, atmospheric mixing ratios of $CO_2$ are set to a constant global value. A simulation with a fully coupled carbon
cycle would have allowed for assessing local interactions between land cover and subsequent $CO_2$ emission changes driven e.g. by the shifts in the Indian monsoon. However, we expect that main features of ocean carbon cycling would remain unaffected.

## 7   Conclusion

By performing experiments, where the sense of rotation of the Earth was changed, we aimed at better understanding climate with respect to hemispheric asymmetries and longitudinal distributions. Two simulations, each encompassing 6990 years
forced by conditions thought to be representative of Earth's climate before the era of industrial combustion (i.e., 1850) are performed: one with a retrograde rotating Earth (RETRO) the other being the control climate (CNTRL) of the prograde rotating Earth. The simulations, while endeavoring to be as comprehensive as possible, are not without limitations. For instance their resolution is coarse, so as to expedite the long-equilibration of the ocean and biogeochemical cycles, and some other constituents (ice-sheets, soil properties, greenhouse gases, aeolian dust input and the atmospheric aerosol) are held constant
consistent with the present day (or pre-industrial) prograde rotating Earth. These simplifications could be relaxed in future

studies. Even so the simulations are the first of their kind as two previous studies looked in a much more limited way, and over shorter timescales, at how the ocean responds to the sense of planetary rotation.

Overall differences in global and zonal mean quantities between RETRO and CNTRL are small, indicating that these quantities are rather robust features of the Earth's climate and do not depend on the sense of rotation. Hemispheric asymmetries consistent in both simulations seem to depend on the land-sea distribution rather than on the atmospheric or oceanic circulation. Longitudinal differences between these experiments, while considerable, are for the most part as one would have expected given a basic understanding of climate physics proposed by von Humboldt and Köppen. The zonal wind patterns reverse in RETRO relative to CNTRL, which results in mid-latitude easterlies instead of westerlies, and tropical westerlies (and trades) instead of easterlies. Due to the reversed winds in RETRO, sub-tropical and mid-latitude continents become colder on their western coasts and warmer in the east. Temperature changes over Eurasia are especially prominent with a strong wintertime cooling over Northwest Europe ($>20$ K). The continental climates show shifts in isotherms consistent with expectations dating back to Alexander von Humboldt's ideas now more than 200 years old. Changes in climate zones are in broad agreement with the predictions made by Wladimir Köppen nearly one century ago. For instance, deserts shift from the western subtropical continental boundaries to the southeast. In the ocean, the western boundary currents show up as eastern boundary currents in RETRO, associated with a strong poleward heat transport in the subtropics, which is also to be expected from classical oceanographic theory. The classical theories explain many features of the climate, as is indicated by their success in predicting many of the simulated climate changes in RETRO.

However, not all of the changes were expected from classical theories, nor necessarily trivial. One of the most unexpected differences is in patterns of precipitation. In the extra tropics storms show greatest track densities over land, rather than over the ocean, even if the intensity and number of storms do not show detectable changes. Likewise there is a northward shift of the extra-tropical storms, and the main regions of baroclinicity. Over North America a major storm track develops to the north of the Great Lakes. Cyclones transport moisture from south Asia and the Indian Ocean towards the Mediterranean and the Atlantic.

In the tropics, the rainbands shift from a double ITCZ centered in and around the warm-pool Asia-Australia monsoon complex in CNTRL, to a more tri-polar structure with centers of action in the eastern tropical Atlantic, the central Pacific and a monsoon region centered in the Middle East. A southward shift in the position of the annually and zonally averaged position of the rain bands is consistent with shifts in the zero-line of the atmospheric energy transport (the intertropical convergence zone). The shift in this line is as would be anticipated by changes in the global climate, i.e., from a warming of the tropics or increased northern hemisphere baroclinicity (Kang et al., 2008; Bischoff and Schneider, 2014). The origin of these changes is, however, not clear as the magnitude and position of the rainband shifts do not agree quantiatively with changes in the energy-flux near the equator. The latter suggests that stationary eddy transport and air-sea interaction (Wallace et al., 1989; Philander et al., 1996) are important for determining the position of the zonally and annually averaged rainbands.

Another unexpected change, albeit one in line with the changes in patterns of precipitation, is a shift in the desert climates from the Northern to the Southern Hemisphere. The simulations predict a complete replacement of the wide desert belt from West Africa to the Middle East in CNTRL by more moderate, humid climates in RETRO. This retreat of desert climates in

Africa and Eurasia is accompanied by an extensive formation of dry climates in South and North America, but not so dry as the African dry climates they replace. Hence, Earth becomes greener in RETRO with more biomass. Changes in biomass, and precipitation patterns are to the first order consistent with the America's adopting the climate pattern of present-day Euro-Africa, and vice versa.

The exchange of Euro-African with American climate, and the shifts in fresh water transport by the atmospheric circulation are accompanied by large-changes in the ocean overturning circulations. An important and outstanding question of physical oceanography is why the North Atlantic is so disproportionate in its production of deep water, with very little or no deep water formed in the North Pacific. Kamphuis et al. (2011) suggested that the more southward continental boundary of the Pacific limits deep water formation as compared to the Atlantic, which extends well into the Arctic. Other studies (e.g., Warren,

1983) have suggested that the freshening of the Pacific relative to the Atlantic stabilizes the overturning pattern. A prominent difference between RETRO and CNTRL is the collapse of the Atlantic MOC in RETRO, with only sporadic formation of deep intermediate water in the North Atlantic. At the same time a strong meridional overturning cell emerges in the Pacific. The Pacific MOC in RETRO is similar in structure but slightly stronger than the Atlantic MOC in CNTRL. The breakdown of the Atlantic MOC and the associated decrease in meridional heat transport leads to a cooling of the North Atlantic associated with

a southward extension of sea ice and a significant cooling over Europe. The accompanying southward shift of the temperature maximum over the tropical Atlantic contributes to the southward shift of the Atlantic ITCZ, through enhanced baroclinicity (and hence heat transport) as argued above. In contrast, the Pacific MOC in RETRO, characterized by an enhanced northward transport of heat along the west coast of North America, results in a significant warming of the North Pacific. The switch of the deep water formation into the North Pacific shows that changes in atmospheric circulation and moisture transport are sufficient

to create a completely reversed conveyor belt circulation. The topographic setting (basin and continent distribution) does not restrict deep water formation to the North Atlantic and Southern Ocean, while preventing it in the North Pacific contradicting findings from Kamphuis et al. (2011).

In the tropics, the Indian Ocean takes over the role of the eastern tropical Pacific with a strong net heat uptake due to strong upwelling. The pattern of biogeochemical tracers are tightly bound to changes in the ocean circulation. Zonal and global

means of biogeochemical features are very similar in CNTRL and RETRO. However, an unforeseen shift in the dominance of cyanobacteria over bulk phytoplankton is found in the northern part of the Indian Ocean. Here, the interplay between strong upwelling and basin geometry leads to the development of an extended oxygen minimum zone with increased denitrification and nutrient trapping. Upwelling of phosphate enriched and nitrate depleted water leads to favourable growth conditions for cyanobacteria.

Changes in the carbon inventory of the terrestrial and the marine realm in RETRO are rather small. The global carbon storage on land increases by 86 PgC; the marine carbon inventory increases by $\sim$100 PgC, both compared to CNTRL. The increase in carbon storage of both subsystems is only possible due to the constant prescribed atmospheric $CO_2$ concentration acting as an unlimited carbon source. With a fully coupled carbon cycle an increase of the land carbon inventory would tend to reduce the atmospheric $CO_2$ which would lead to a compensating effect (outgassing) by the ocean. The projected change on land

(increase by $\sim$86 PgC) is rather small compared to the marine carbon inventory ($\sim$38000 PgC). When taking characteristics of

the marine carbon chemistry and the long simulation period into account, the ocean is capable to compensate for most (more than 80%, Archer et al. (2009)) of the additional carbon stored on land in RETRO. Thus, we expect the effective change of the atmospheric $CO_2$ in a fully coupled simulation to be smaller than 7-8 ppmV. Such a change in atmospheric $CO_2$ would have only minor effects on climate.

Sensitivity studies suggest that differences in the circulation in the RETRO versus CNTRL simulations do not fundamentally affect Earth's climate sensitivity compared to e.g. the effect of model resolution. Changes in the ocean in RETRO do, however, appear to affect the time-scale of equilibration.

     Overall changes in the interaction between the atmosphere and the ocean that accompany a reversal of the sense of planetary rotation help demonstrate the important role of the ocean in climate and provide valuable out-of-sample tests of the robustness

of climate models and the physics governing their response to perturbations. Repeating the RETRO/CNTRL pair of simulations at higher resolutions and with other models could thereby provide a fascinating laboratory for testing understanding of climate models and, by inference, of Earth's climate.

## 8    Code availability

MPI-ESM is available under the Software License Agreement version 2, after acceptance of a licence

(https://www.mpimet.mpg.de/en/science/models/license/).

## 9    Data availability

Data are available from https://cera-www.dkrz.de/WDCC/ui/cerasearch/entry?acronym=DKRZ_LTA_110_ds00001.

## Appendix A:  Changes in the El Niño Southern Oscillation

The analysis of the geographical distribution of the climate mean surface temperature (Figure 2, described in Section 3) is

extended by a variability analysis focusing on the tropical Indian-Pacific Ocean (40°E–60°W, 30°S–30°N) to capture ENSO-related year-to-year fluctuations (deviations from the climate mean, Figure A1) for RETRO and CNTRL.

     Figure A1 shows the two leading EOFs which, covering the area from the East African to the South American coast (from (40°E–60°W, 30°S–30°N. The two leading EOFs explain 36% and 14% of the total variance for RETRO (62% and 10% for CNTRL). The dominating variability is centered in the Indian Ocean (tropical Pacific for CNTRL). The second EOF captures

variability in the western tropical Pacific.

     The sectional seasonality of the sea surface temperature anomaly (SSTA, averaged between 5°S and 5°N) is diagnosed in terms of the monthly standard deviation after removing the climatological annual cycle (Figure A2). In CNTRL, the strongest variability, which occurs in the eastern tropical Pacific in July-August, is different from the observed ENSO peaking in boreal winter months, but the weak variability that characterizes the spring predictability barrier is clearly seen (Figure A2). In

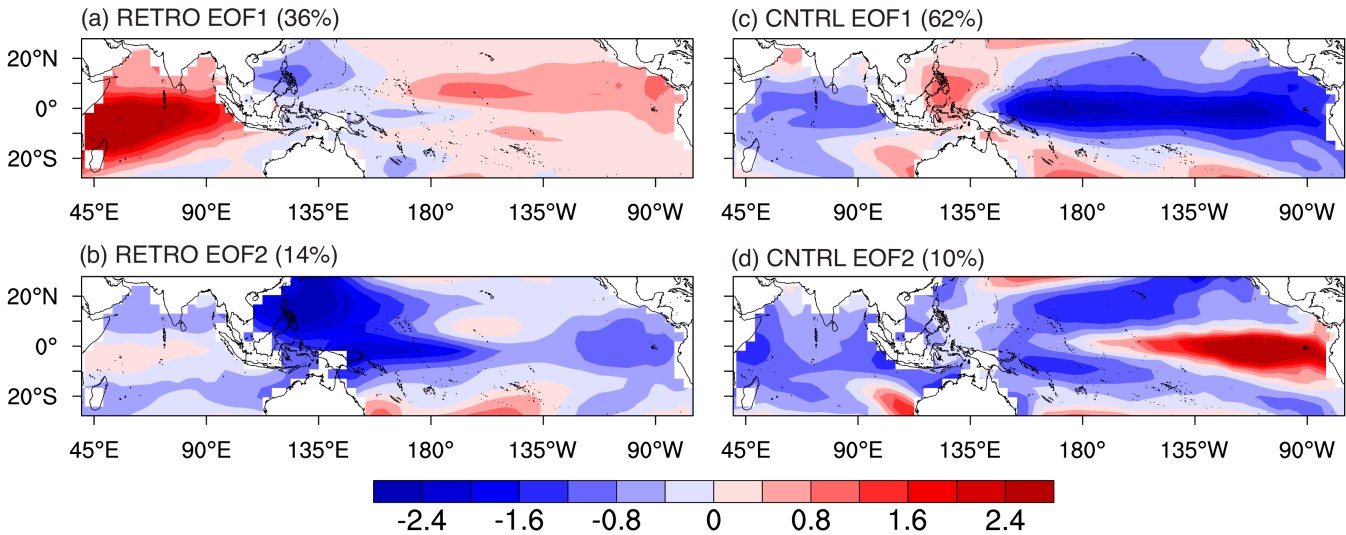

**Figure A1.** Two leading EOFs of yearly SST Anomaly (40°E–60°W, 30°S–30°N): RETRO (left) and CNTRL (right)

RETRO, the center of strong variability shifts to the Indian ocean and to the middle tropical Pacific with peaks in May-July and around October-November, respectively.

The seasonal cycle of the variability of the indices characterizing only the Indian ocean INDINO (10°S to 0°S for RETRO) and the tropical Pacific Nino3/4 (for CNTRL) as well as the spectral densities of these indices are shown in Figure A3. While the Nino3/4 index reveals a broad peak in the one to ten years period domain, the lower frequency variability in the INDINO index levels off as white noise. Additional diagnostics on the physical causes of the variability behavior and the atmospheric long-distance teleconnections are beyond the scope of this analysis.

## Appendix B:  Influence of the direction of the movement of the sun

Presumably, climate changes simulated in RETRO are due to the differences arising from differences in the sign of the planetary vorticity, as imparted by the Coriolis force acting on the wind. But changing the sense of planetary rotation also changes the diurnal march of the land under the sun. To understand the extent these thermodynamic effects influence the differences between RETRO and CNTRL, one additional 6990 year long simulation (RETRO-S) was performed without changing the sense of diurnal march in insolation. Thus all changes in this simulation relative to CNTRL are due to the reversal of the sign of the Coriolis parameter. Comparison with RETRO isolates the effect of a reversed path of the surface under the sun. Differences between RETRO-S and RETRO are for the most part negligible, and only evident in the distribution of precipitation. Over Africa RETRO-S precipitates $0.5\,\mathrm{mm\,d^{-1}}$ less than RETRO (see Fig. B1), a difference that is in the order of the year-to-year variability, as measured by the standard deviation of annual mean precipitation. Likely as a consequence of this change, RETRO-S also exhibits a clear northward shift of the ITCZ over the tropical Atlantic Ocean.

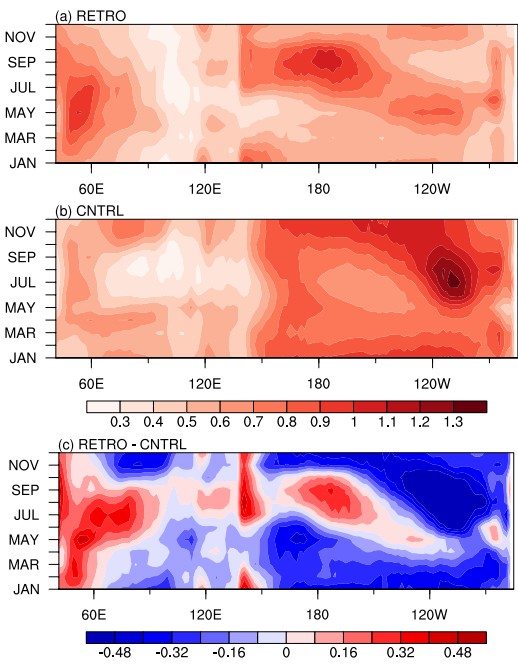

**Figure A2.** Sectional view of the seasonality of the sea surface temperature anomaly (SSTA) at the equator (averaged between 5°S and 5°N) diagnosed as monthly deviation from the climatological annual cycle: a) RETRO, b) CNTRL, c) RETRO–CNTRL

To understand this difference, we look at the diurnal cycle of precipitation (not shown) over the impacted land masses. In agreement with the sun movement, precipitation starts falling on the eastern side of Africa in RETRO-S and moves inland. The opposite is true in RETRO, so that precipitation starts falling with a 2 h to 4 h time lag on the eastern side of Africa (note that the output frequency is 2 h). Despite the later precipitation start, RETRO produces more precipitation. This may be related to the fact that most precipitation falls on the eastern side of Africa in the mean (see Fig. 1), with a background flow over the continent that is directed eastwards. Hence in RETRO, the propagation of the convection due to the sun movement and the background flow work together, whereas in RETRO-S they are opposed to each other. The complementary situation happens on the western side of South America. Here the background flow is westward, which should lead to an enhancement of precipitation in RETRO-S, which is indeed the case, as shown in Fig. B1.

## Appendix C: Climate sensitivity

If climate sensitivity, the longterm response to increased atmospheric $CO_2$, did not depend on the details of the circulation, we would expect it not to depend on the direction of the planetary rotation. To investigate this, we conduct two experiments with

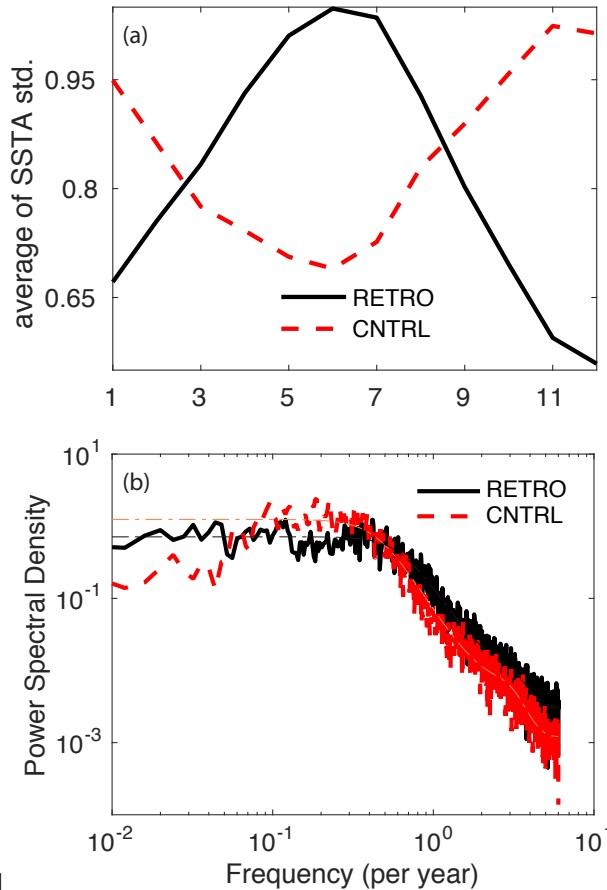

]

**Figure A3.** SSTA indices: (a) seasonality of NINO3.4 (CNTRL, black) and INDINO (RETRO, averaged over 50–70°E, 10–0°S, red) and (b) the respective power spectra.

abrupt quadruplings of $CO_2$ (RETROx4 and CNTRLx4, Figure C1). Here, the equilibrium response is estimated using linear regression on annual means of the relationship between radiation balance and global warming. The slope in such a diagram is thought of as a feedback parameter, $\lambda$, which must be negative to yield a stable climate. We find, indeed, that the climate sensitivity is not very different between CNTRLx4 and RETROx4, roughly 10 percent larger in the latter (4.00 vs. 4.45 K

5 per doubling of $CO_2$). This is substantially less than the difference to the slightly higher resolution MPI-ESM1.2-LR model (also shown, 3.19 K) and much less than the spread among CMIP5 climate models (2.0-4.6 K) and scientific uncertainty as expressed by the likely range from the IPCC AR5 (1.5-4.5 K). It should be noted that the here reported climate sensitivities are regressed over the years 21-1000. Using instead the more standard regression over years 1-150 we obtain 3.46, 3.92 and 2.87 K for CNTRLx4, RETROx4 and MPI-ESM1.2-LR.

10 The small increase in the longterm feedback parameter for RETROx4 of about $\delta\lambda \approx +0.09\,\mathrm{W\,m^{-2}\,K^{-1}}$ stems primarily from cloud shortwave effects. Indeed, it was tempting to think that the more widespread sea ice cover in the North Atlantic

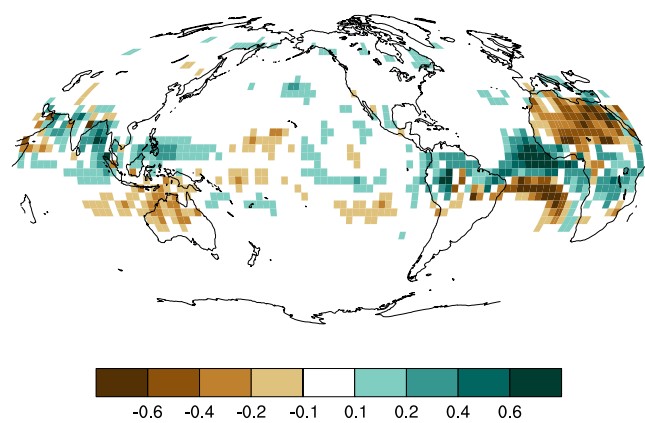

**Figure B1.** Mean precipitation difference [mm day$^{-1}$] between RETRO-S and RETRO. Note the different color scale when comparing to Fig. 2.

in RETRO would cause a stronger surface albedo feedback, but the clear-sky shortwave component does not differ between the simulations. Likewise, the longwave feedbacks differ by merely 0.01 W m$^{-2}$ K$^{-1}$. Thus, the remaining 0.08 W m$^{-2}$ K$^{-1}$ stems from cloud shortwave feedbacks.

The difference in feedback parameter between the earlier parts of the simulation (years 1-20) and later parts (years 21-1000)
is thought to be due to tropical cloud feedbacks (Block and Mauritsen, 2013; Andrews et al., 2015; Zhou et al., 2016). One idea is that surface temperatures in the ocean upwelling regions in the tropical East Pacific warm slower than the surface waters in the Warm Pool region, causing more low-level cloudiness in the East as a result of increasing low-level stratification. The striking similarity between CNTRx4L and RETROx4 is suggestive that the details of the tropical atmosphere and ocean circulations also do not matter for determining the strength of this mechanism.

Finally, we noted a striking difference between CNTRLx4 and RETROx4 in the longterm equilibration for years 1001-2500. As found for an earlier version of the model by Li et al. (2013), CNTRLx4 exhibits a third time-scale where $\lambda$ becomes more negative causing the system to equilibrate faster, whereas, if anything, RETROx4 does the opposite. The third timescale found in CNTRL and in the ECHAM5/MPIOM model used by Li et al. (2013) is yet to be found in other models partly because models are only seldom run long enough. Previously, we thought the behavior originated in the coarse resolutions used in these
runs, as the MPI-ESM1.2-LR exhibits a faster equilibration, but the contrasting very slow equilibration of the coarse resolution RETRO simulation could indicate that the properties of the ocean circulation are indeed important for the equilibration process.

**Appendix D: Iron supply in marine biogeochemistry**

To assess the contribution of aeolian iron input into the Northern Indian Ocean to the growth of cyanobacteria we estimated the minimal iron flux into the upper 55 m due to upwelling (Fig. D1). One of the characteristics of the marine iron cycle is that
iron concentrations in the deep ocean are almost constant (0.6 – 0.8 $\mu$mol Fe m$^{-3}$ for Atlantic and Pacific, Moore and Braucher

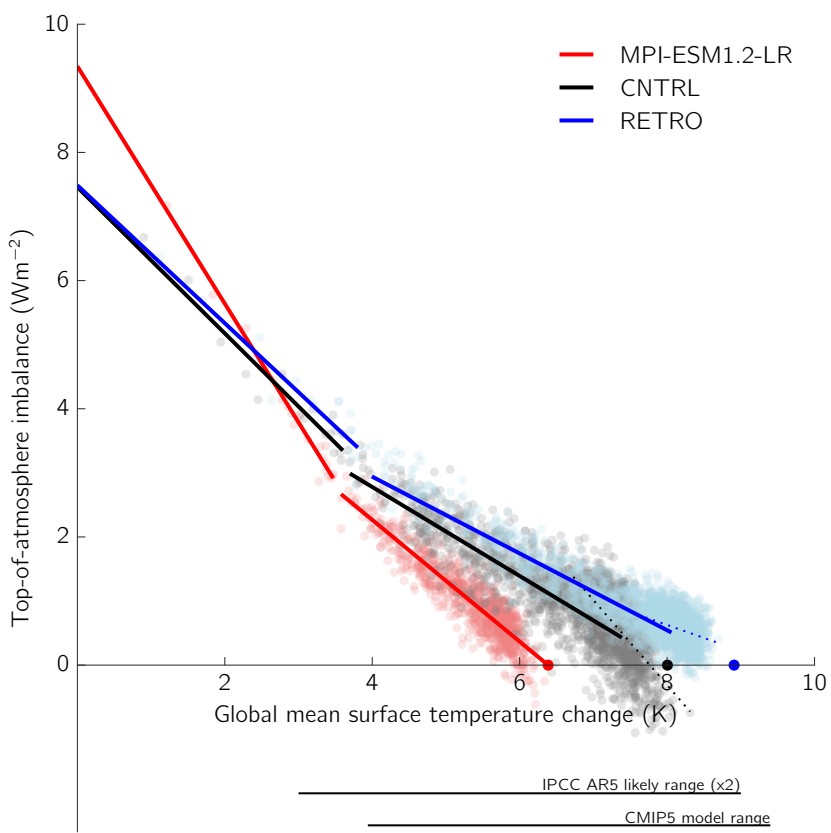

**Figure C1.** Surface temperature change versus top of the atmosphere energy imbalance for RETROx4 and CNTRLx4 each run for 2500 years following an abrupt quadrupling of atmospheric $CO_2$. Small dots are annual means. For comparison a 1000-year run with MPI-ESM1.2-LR, i.e. a higher resolution version than CNTRL, is shown. The lines are linear regressions over years 1–20 and 21–1000, respectively, and the solid dots on the x-axis shows the linear extrapolation to stationarity based on the year 21-1000 regressions. Also shown for CNTRLx4 and RETROx4 as dotted lines are the regressions for the years 1001–2500. The lines in the lower part of the figure show the IPCC AR5 assessed likely ($p > 0.66$) range as well as the range of responses found in CMIP5 climate models.

(2008)). When multiplying the upwelling velocity at 55 m water depth of RETRO with this background concentration (set to 0.6 $\mu\mathrm{mol\,Fe\,m^{-3}}$ in HAMOCC) we achieve the lowest estimate of the iron supply due to circulation. For comparison, we calculate the iron demand for net primary production of bulk phytoplankton and cyanobacteria. As both phytoplankton types contain carbon and iron at the same ratio in our model, the latitudinal structure of the zonal mean iron demand is identical to

that of the zonal mean net primary production shown in Fig. 17. Even this lower bound estimate of the iron upwelling flux satisfies 40-80 % of the iron demand of cyanobacteria in the Northern Indian Ocean. It is likely that sea water concentrations of iron in the upwelling waters are higher than the background value even without local aeolian input. The cause for this is the interplay between the high primary production/remineralization at the Equator, which leads to an accumulation of nutrients (phosphate and iron) in the upper ocean and the meridional overturning circulation (Fig. 17) that creates a northward transport.

The lack of nitrate (Fig. 18) in these upwelling waters inhibits growth of bulk phytoplankton fostering the local dominance of cyanobacteria.

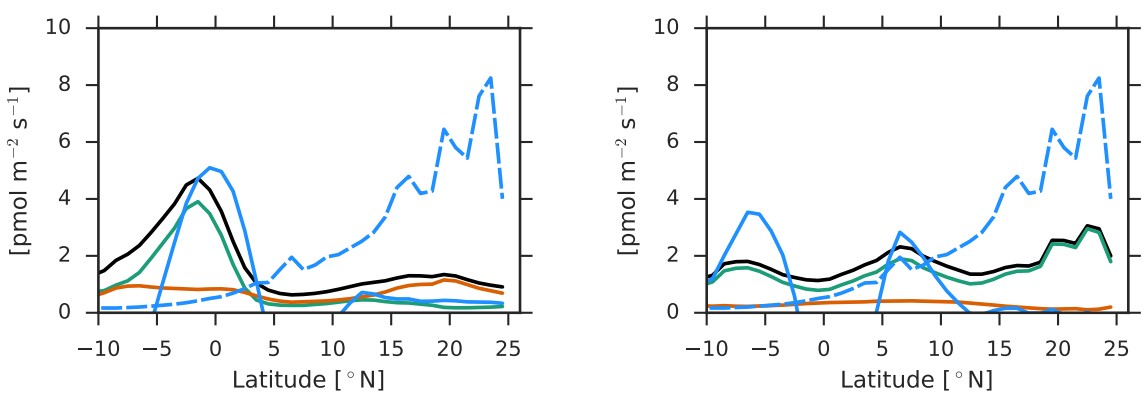

**Figure D1.** Zonal means in the Indian Ocean for RETRO (left) and CNTRL (right) of iron demand for net primary production (all phytoplankton: black, bulk phytoplankton: green, cyanobacteria: red), aeolian iron input (dashed blue), and iron flux due to upwelling (solid blue). All fluxes are given in [pmol Fe $\mathrm{m^{-2}\,s^{-1}}$].

## Appendix E: Storm track changes

In this section, additional results expanding on the tracking of cyclones in the northern hemisphere during boreal winter (DJF, section 3), as well as the analysis for the southern hemisphere during austral winter (JJA) are presented. Figure E1 shows the

zonal averaged statistics of cyclones tracked in the northern hemisphere. Results show that, while the storm tracks shift from the ocean to the land (Fig. 4), there is little difference in the zonal mean statistics between RETRO and CNTRL besides of the baroclinic growth rate.

In the southern hemisphere, cyclones shift from the Indian and Atlantic ocean to the Pacific ocean, as presented in Fig. E2. Consistent with that, the cyclones in the Pacific Ocean in the retrograde planet are stronger, have higher growth rates, and are

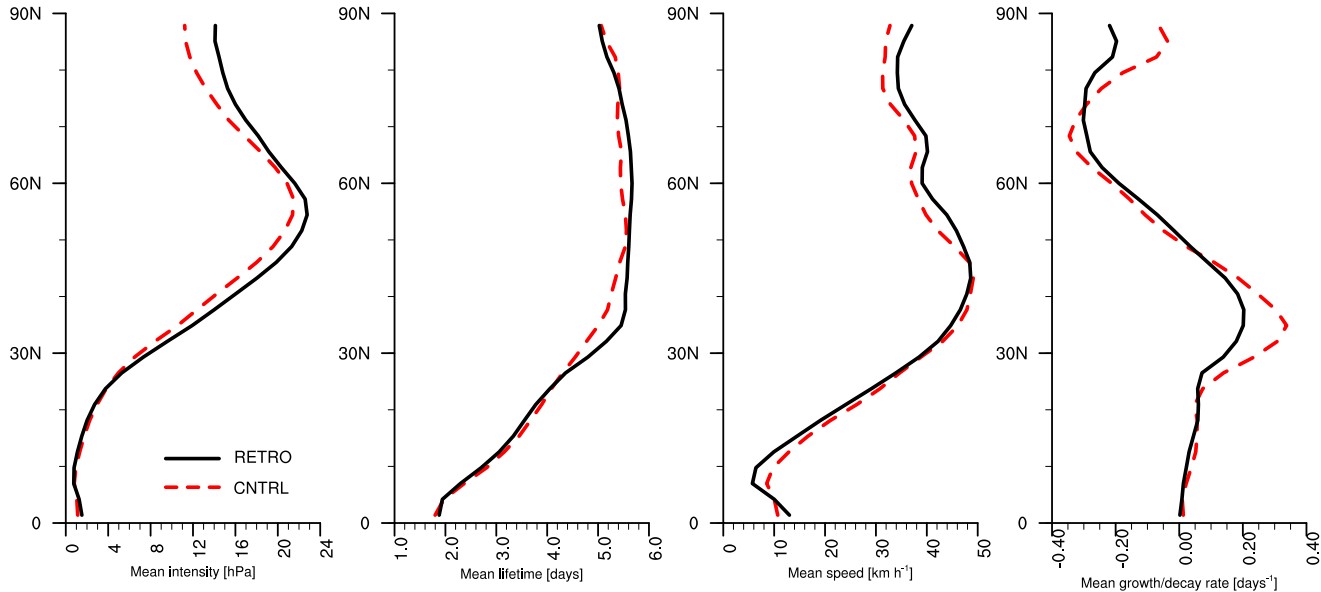

**Figure E1.** Zonal averaged statistics obtained from the tracking algorithm for the DJF season in the northern hemisphere.

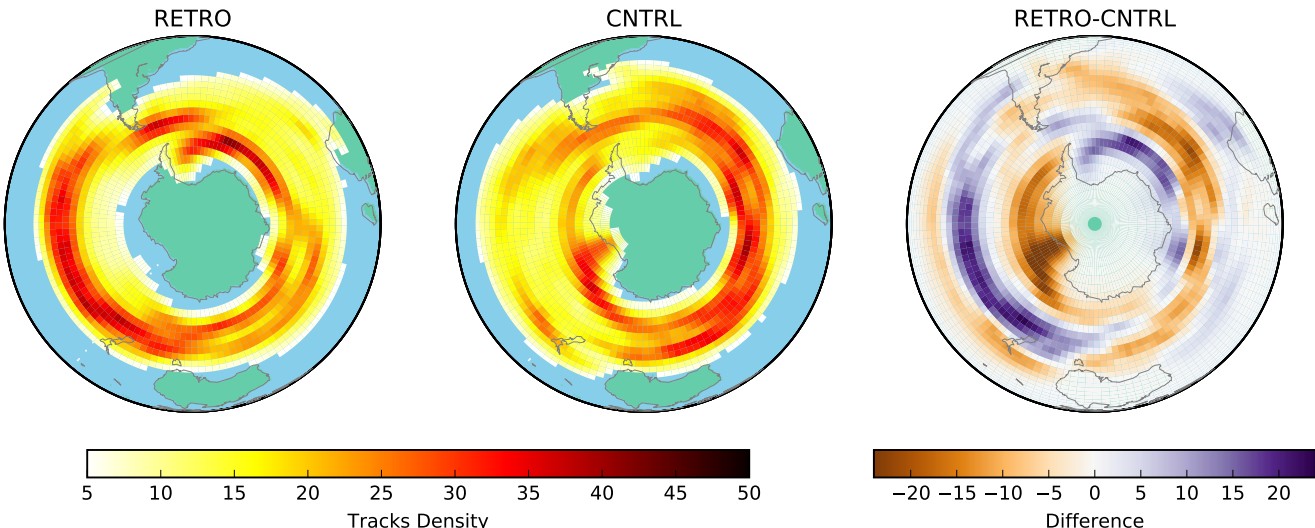

**Figure E2.** As in Fig. 4 but for the southern hemisphere during the JJA season.

also faster (Fig. E3). However, these local changes are compensated such that the overall zonally-averaged statistics in the southern hemisphere do not change significantly (not shown).

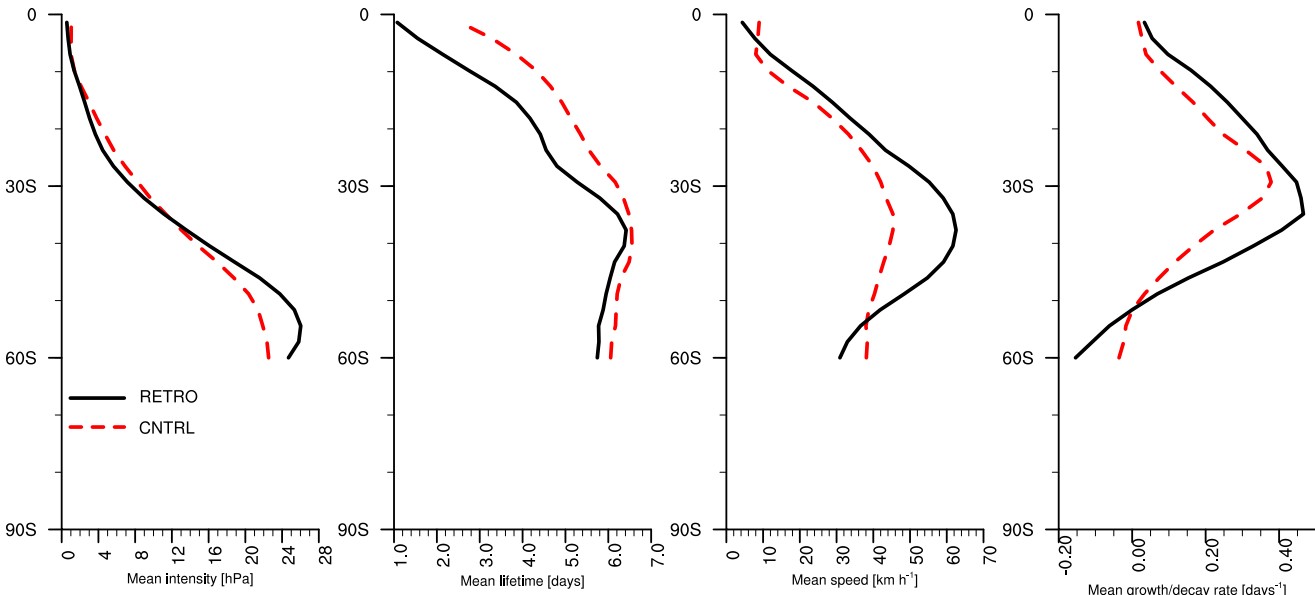

**Figure E3.** Zonal averaged statistics obtained from the tracking algorithm for the JJA season in the southern hemisphere Pacific Ocean region.

*Competing interests.* The authors declare that they have no conflict of interest.

*Acknowledgements.* This work was in part supported by German Federal Ministry of Education and Research (BMBF) as Research for Sustainability initiative (FONA); through the project PalMod (FKZ: 01LP1502A and 01LP1504C). All simulations were performed at the German Climate Computing Center (DKRZ). Norbert Noreiks created the Köppen turnip plot in Fig. 9. Finally, we thank two anonymous reviewers and Daniel Kirk-Davidoff, who helped to significantly improve the manuscript.

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
