# Peer review of "The climate of a retrograde rotating earth"

_Earth System Dynamics, 2018_

## Referee Comment (RC1) · Anonymous Referee #1 · 25 Jun 2018

Mikolajewicz and coauthors use the coarse resolution version of the Max Planck Institute Earth System Model to simulate a retrograde Earth, in which the sign of the Coriolis force is changed and the direction of the suns diurnal march is reversed (RETRO) and compare to an extension of a Pre-industrial control simulation developed for CMIP6. Some caveats include the use of PI fixed aerosol loading, GHG concentrations and ice sheet distribution.

I thought the manuscript was scientifically interesting and well written, however I had a couple comments/concerns with the analysis and therefore request a minor revision. Please see below.

Major comments:

[Figure]

I thought the expansion of sea-ice in the North Atlantic in the RETRO experiment was an interesting result that the authors link to the collapse of the AMOC and the associated reduction in northward heat transport. I also wondered if it was partly due to the reversal of the subpolar gyre and advection of colder Arctic air/water southward into the North Atlantic? It would be nice to plot the wind stress vectors on the sea ice images.

Does your thermodynamic sea ice model take into account brine rejection? This would surely impact the density (making it saltier and denser) of the North Atlantic circulation. If not, this would be a caveat of the current study and one possible theory for the discrepancy between this study and the Kamphuis et al. study that used the CCSM3 model, which I believe does reject salt, in which the Atlantic MOC did not shut down.

On page 17 you mention that both the Subtropical and Subpolar Gyres in the North Atlantic are weaker in the RETRO simulation and that it is a consequence of a weaker thermohaline circulation. Can you explain? I think of the gyres as being predominately wind driven. If there are weaker Atlantic gyres, then I assume net evaporation decreases substantially in the RETRO case compared to the control, which could be one of the drivers of the collapsed Atlantic overturning in RETRO. On this note, it would be very useful to show not only the mean precipitation and the difference but also Evaporation minus Precipitation and the difference.

Although the strength of the Pacific overturning circulation in RETRO is much stronger than the Atlantic overturning in the CNTRL simulation, the global net oceanic heat transport (Fig. 13a) shows less OHT in the RETRO compared to CNTRL. I expected a stronger MOC to result in more northward heat transport. Can you comment on this apparent discrepancy?

The dominance of cyanobacteria in the North Indian Ocean is one of the main conclusions of the paper, as mentioned in the abstract, however I think there is too much uncertainty in this conclusion since the dust loading is fixed in both simulations. Although the authors claim the change in upwelling rates is the driver, they write that cyanobacte-

ria is able to grow as long as there is sufficient phosphate and iron availability. I think a sensitivity test where you remove dust loading in the RETRO experiment is necessary to show that it is indeed a result of the increase in upwelled phosphate.

I would move the storm track analysis to the main paper – not in the appendix. I thought this was useful in understanding the precipitation changes in figure 2.

Technical comments:

Abstract: There are two instances of the word surprising. I would use a different word: unexpected?

Page 2 line 31: combine should be combined

Page 2 line 32: to provide should be provides

Page 3 line 8: around Africa

Page 3 line 16: limited northward extension (of what), the gyre circulation? I would be more specific.

Page 4 line 4: of which aspects

Page 4 line 13: vertical

Page 9 line 1: You write that cyclogenesis is shifted equatorward in RETRO. I don't see this in Figure 2d, e. Do you mean over the South Pacific? It seems the precipitation increase over western South America is related to the anomalous onshore flow and its interaction with topography.

Page 12 line 11: variety is spelled wrong

Page 17 line 4: what used to be

Page 17 line 14: the sentence structure is awkward here.

Page 20: In the caption to figure 13, Indian is spell wrong. Also, why not show the

global freshwater transport (black line) in panel b?

---

## Referee Comment (RC2) · Anonymous Referee #2 · 4 Jul 2018

*General Comments

The authors use a Earth system general circulation model allowing considerable process complexity (although at a lower spatial resolution than the current state of the art) to simulate the climate of a counterfactual Earth that rotates around its axis in the opposite direction. The study is fascinating in conception, and appears to have been carried out and presented in a clear, careful and scientific manner. I have some comments and queries on specific matters of course, but the paper is sound and could really be published with only minor revision.

*Specific Comments

The biggest single improvement, in my opinion, would come from the authors being

little clearer on the intended aims and conclusions from the study. Given our imperfect understanding of and ability to simulate the world as we see it now, it is obviously a bit of a stretch to apply precisely the same tools to a world for which we have no observational verification or ability to conclusively falsify hypotheses. I don't think that means that we shouldn't carry out an exercise like the one that has been presented here, but having a very clear framework in place, eg "we want to test our general understanding of a certain feature of the real Earth", or "we're simply interested in exploring what such a place would look like for its own sake" - both would be quite valid, as far as I'm concerned - is helpful in interpreting what is shown. The current manuscript feels like it tries to do both at once without clearly framing the context for some of the results, and I'm left unsure, for instance, whether certain aspects of the write-up are really meaningful - one's interpretation depends on the context in which you read them.

For instance, in the RETRO-S experiment, the Coriolis force is reversed but the direction of the diurnal solar cycle is left alone. The configuration is totally artificial and the physics in question is almost certainly not being reproduced "realistically" (even very high resolution NWP models have difficulty getting the physics involved in the daily timings of the afternoon rain in Africa right), so it's questionable that this has direct relevance to learning about our world. But it's an interesting, physically valid part of the artificial construct we're being presented with when seen outside of a real-world context - I think the same sort of comment applies to the climate sensitivity section. The opposite is however true for some of the land-cover and ocean biogeochemistry results, where the RETRO model is tied to real-world distributions of $CO_2$, soil, ice and dust and it's difficult to see what results as being internally consistent in the context of a backwards-rotating ecosystem.

A symptom of this may be that almost the entirety of the Conclusion section is a descriptive summary of what has been seen (and already described) in the simulation, rather than making any attempt to help the reader see what has been learned from the

exercise in toto. Apart from having certain aspects of this fascinating Gedankenexperiment demonstrated for real, I'll admit to personally being unsure what robust conclusion might be taken from this (other than that playing with climate models is fun), so any help the authors could offer would be appreciated.

pg1, paragraph 2: some of the summarised effects and causes in the abstract don't come out very clearly in the main body of the manuscript itself, requiring the reader to piece together material in the different sections and appendices for themselves to make a picture that matches up with the abstract. Some of this could be explicitly outlined in the Conclusions section

pg2, line 29: the literature on monsoons is vast, I find it difficult to believe there is only one study looking at what sets their locations

p3,l27: Does "These simulations did not [...]" refer to just the simulations of Kamphuis et al, or of the results of both Smith and Kamphuis taken together?

p3,l34: "for the most part" is very imprecise

p4,l13: it would be helpful to say where the atmosphere model top is, in terms of how much of the stratosphere is being modelled.

p4,l[20|26]: is the "long" model spinup (l20) different from the 6990 year model integration (l26), and if so, how long was it?

p4,l30: how one interprets the impact of not allowing ice, GHGs and aerosols to change very much depends on how you frame the results of the simulation - see above. The significant changes simulated in biological production and ocean deepwater would be expected to have an impact on the overal GHG concentration in the atmosphere, with a possible first order effect on the climate and possible major ice-sheet feedback - doing this sort of thing with enough detail is clearly out of scope for our present modelling capability, but it would be fun and informative to have some speculation from the authors!

p5,l3: piControl needs explaining

p5,l4: has ESM been defined yet?

p6,l16: why is the Pacific cold tongue so much less evident in RETRO?

p18,l29: could the authors speculate from the published model specifications and results why each model sees the MOC changes it does?

p21-24. Section 6 describes one of the most interesting aspect of the results and the importance of using an ESM, rather than a purely physical climate model, to do this kind of simulation. It's a real shame that the connections aren't there to either include the impact of changes in landcover and dust on the biogeochemistry, nor the follow on impact on ocean CO2 sequestration of the biogeochemistry and physical ocean changes together. The authors end by claiming that they expect that their model limitations don't affect the main features of the ocean carbon cycle, but I'm much less sure, and I'd welcome either more justification of this statement, or some intersting speculation on what could happen instead.

p27. Section 7 feels unfocussed, and (given the main "framing" comment above) I'm really not sure what one is supposed to take from it

p28,l7: I don't think "numerous" sensitivity experiments have been shown, just 2: RETRO-S and a 4xCO2 run?

p28,l12: I also don't think one can credibly call ice-sheets, soils and GHGs "minor" consituents of the Earth system!

p30, Appendices: In my opinion some of the most interesting analyses have been relegated to relatively brief descriptions in the appendices. Personally, I would rather see all of this material in the main text in preference to the current section 7, although some of it may be a bit specialised and the Editor may want to take a view based on the intended audience for the journal. The other issue would be whether the features are really modelled robustly enough to be part of the main attraction - the functioning of an ENSO and its teleconnections on a RETRO planet could probably be a paper all on

its own rather than 18 lines in an appendix, if the relevant physics in the model could be shown to be robust enough.

p34,l8: How many years of data have been used in the power spectrum? The confidence intervals don't look very convincing - if the 1-10yr signal in CNTRL is supposed to look significant then so is 10+ in RETRO...?

*Technical Corrections

The authors seem to have a habit of trying to make sentences out of phrase fragments that have no main clause eg pg3, line19 "One which includes different net freshwater forcing regimes for the ocean, while keeping the present continental geometry.", p6,l34 "The area with the largest amount of annual precipitation being near Ascension Island in the Southern Tropical Atlantic (...).". There are several occurrences of this construction throughout.

Several features are described as (objectively) "surprising" or "(un)expected", which are subjective assessments

pg2, line27: "that what would be caused" - what should be which

p3,l5: "how the ocean transport heats"

p3,l[11|15]: "stabeliz[es|ing]"

p3,l25: "FAMOUS" needs an explanation/citation if you're going to specify the model

fig 1. why does the time axis go from 1000-8000 for runs that are described as 6990 years?

p4,l7: "MPI-ESM" and "CMIP6" need explanation/citation

p4l13: "vertic"

p5,l12: "mid-altitudes have surface easterlies

p6,l2: "feint" should be "faint"

p6,l16: "the American's warm"

p6,l34: "like presentday Palau"

fig3: "(10 Mm versus 8Mm)" I don't understand this part of the caption

fig4: whilst very pretty, this figure lacks adequate labels as to what the flux components are, what time averaging has been applied, and takes up a lot of space illustrating something that "hardly differs from that of the CNTRL" and (pg9,l10) whose quantities might be expected to vary most interestingly on a hemispheric basis, rather than the global average actually shown

p11,l3: new paragraph at "Changing the planetary rotation [...]"

p12,l11: "vareity"

fig11,13: "Indic" Indian Ocean?

p21,l27: "[...] occur only very localized."

p24,l10: Despite the fixed atmospheric $CO_2$, the changes in the land biosphere in section 4 are extrapolated to an estimate of their potential impact on the $CO_2$ concentration in the atmosphere - could a similar estimate not be made from the changes in the ocean carbon inventory?

p29,l16: "Euro-Africa, with American" doesn't need the comma

p29,l20: "Other studies" needs references

p33,l1: "Kongo"

p33-35: figs C2,C4 not actually referred to in text.

p34,l4: "In RETRO" is repeated in this sentence

p37,l13: "n/a-n/a" in citation

p38,l[25|32]: two web links in citation

p39,l11: two web links in citation

---

## Author Comment (AC1) · 7 Aug 2018

please see the attached pdf

Please also note the supplement to this comment:
https://www.earth-syst-dynam-discuss.net/esd-2018-31/esd-2018-31-AC1-supplement.pdf

---

## Author Response (AR1)

**The climate of a retrograde rotating Earth**
**Response to Reviews**

Uwe Mikolajewicz and Co-authors

August 2018

**Reply to Reviewer 1**

Mikolajewicz and coauthors use the coarse resolution version of the Max Planck Institute Earth System Model to simulate a retrograde Earth, in which the sign of the Coriolis force is changed and the direction of the suns diurnal march is reversed (RETRO) and compare to an extension of a Pre-industrial control simulation developed for CMIP6. Some caveats include the use of PI fixed aerosol loading, GHG concentrations and ice sheet distribution. I thought the manuscript was scientifically interesting and well written, however I had a couple comments/concerns with the analysis and therefore request a minor revision. Please see below.

We thank the reviewer for the positive feedback and for the helpful and constructive comments. A point-to-point reply is given below.

I thought the expansion of sea-ice in the North Atlantic in the RETRO experiment was an interesting result that the authors link to the collapse of the AMOC and the associated reduction in northward heat transport. I also wondered if it was partly due to the reversal of the subpolar gyre and advection of colder Arctic air/water southward into the North Atlantic?

The general expansion of Arctic/North Atlantic sea ice is a typical feature seen in model runs with collapsing AMOC. It is rather the reversal of the gyre that is responsible for the stronger sea ice expansion in front of Ireland in comparison to Newfoundland.

It would be nice to plot the wind stress vectors on the sea ice images.

We included vectors of sea ice transport. Illustrates nicely the answer to the previous point.

Does your thermodynamic sea ice model take into account brine rejection? This would surely impact the density (making it saltier and denser) of the North Atlantic circulation. If not, this would be a caveat of the current study and one possible theory for the discrepancy between this study and the Kamphuis et al. study that used the CCSM3 model, which I believe does reject salt, in which the Atlantic MOC did not shut down.

Also in our model brine is rejected when sea ice is formed. In the model description we have added a reference to a paper, in which the sea ice model is

described.

*On page 17 you mention that both the Subtropical and Subpolar Gyres in the North Atlantic are weaker in the RETRO simulation and that it is a consequence of a weaker thermohaline circulation. Can you explain? I think of the gyres as being predominately wind driven.*

Present days North Atlantic gyres have a wind driven component, but include also 15-20 Sv from the upper branch of the AMOC. This water moves northward in the western and northern margin of the subtropical gyre. The subpolar gyre intensifies in general with more deepwater formation around southern Greenland. It has been a while, that at MPI the stability of the AMOC was investigated in uncoupled ocean model runs with fixed winds. But in these experients it was quite a common feature to see this effect happen. As an example we suggest to look at plate 1 of Maier-Reimer, Mikolajewicz and Crowley, 1990, Paleoceanography 5,3,349-366, which contrasts the surface velocity field of runs with and without a collapsed AMOC. We tried to make this slightly more clear in the text, however, without adding a detailed discussion.

*If there are weaker Atlantic gyres, then I assume net evaporation decreases substantially in the RETRO case compared to the control, which could be one of the drivers of the collapsed Atlantic overturning in RETRO. On this note, it would be very useful to show not only the mean precipitation and the difference but also Evaporation minus Precipitation and the difference.*

Evaporation in the North Atlantic indeed decreases in RETRO. Main reason are the colder SSTs due to the reduced Atlantic heat transport. The opposite is true in the Pacific. The relevant quantity for the ocean are not local net freshwater fluxes, which would be dominated by the longitudinal signal, but the integrals over certain basins or certain latitude bands thereof. Therefore we believe, that fig. 15b is sufficient to extract the relevant information. We believe, that an additional figure with 3 panels would add only relatively little information. The precipitation plot itself is essential for the atmospheric and terrestrial parts of the paper.

*Although the strength of the Pacific overturning circulation in RETRO is much stronger than the Atlantic overturning in the CNTRL simulation, the global net oceanic heat transport (Fig. 13a) shows less OHT in the RETRO compared to CNTRL. I expected a stronger MOC to result in more northward heat transport. Can you comment on this apparent discrepancy?*

The NPDW in RETRO is slightly warmer than the NADW in CNTRL, resulting in a weaker heat transport than would be expected by just comparing the strengths of the overturning. We now mention in the text the relative differences in water mass properties between NPDW in RETRO and NADW in CNTRL.

*The dominance of cyanobacteria in the North Indian Ocean is one of the main conclusions of the paper, as mentioned in the abstract, however I think there is too much uncertainty in this conclusion since the dust loading is fixed in both simulations. Although the authors claim the change in upwelling rates is the driver, they write that cyanobacteria is able to grow as long as there is sufficient phosphate and iron availability. I think a sensitivity test where you*

remove dust loading in the RETRO experiment is necessary to show that it is indeed a result of the increase in upwelled phosphate.

We are in line with the reviewer that the fixed dust field (or more precisely the aeolian iron input) in both simulations adds some uncertainty to the RETRO results in biogeochemistry – as already noted in the paper. Although with the switch between the role of Euro-Africa and the Americas combined with the reversed sense of the winds having the Atlantic as the dustier basin might not be as wrong as it would seem. In any case, tiven that iron is a necessary micronutrient for all primary producers a complete shutdown of the aeolian iron input is not possible as suggested. However, a back of the envelop calculation, which is outlined in a new Appendix D, might support that the fixed iron input field is of minor importance for the dominance of cyanobacteria in the Northern Indian Ocean. The comparison of cyanobacteria iron demand with the minimum estimate of iron supply by upwelling shows that 40-80 % the iron need are covered by upward transport.

I would move the storm track analysis to the main paper not in the appendix. I thought this was useful in understanding the precipitation changes in figure 2.

We followed the reviewers advice and moved key results from the storm track and moisture transport analysis to the main document.

Abstract: There are two instances of the word surprising. I would use a different word: unexpected?

done

Page 2 line 31: combine should be combined

corrected

Page 2 line 32: to provide should be provides

changed in asymmetries ... provide

Page 3 line 8: around Africa

corrected

Page 3 line 16: limited northward extension (of what), the gyre circulation? I would be more specific.

The northward extension of the basin. We have tried to make the sentence more clear.

Page 4 line 4: of which aspects

That will depend on the outcome from simulations with other models. One example would be, that there appears in all 3 hitherto existing studies both a weakening of NADW formation and intensified formation of intermediate/deep water in the North Pacific. Thus the direction of the changes is the same, however 2 out of 3 models cross a threshold, the last one not.

Page 4 line 13: vertical

done.

Page 9 line 1: You write that cyclogenesis is shifted equatorward in RETRO. I don't see this in Figure 2d, e. Do you mean over the South Pacific? It seems the precipitation increase over western South America is related to the anomalous onshore flow and its interaction with topography.

Looking at this again, with the reviewers comment in mind, we agree that this effect depended too much on what specific feature one looked at, so we

removed reference to this point.

corrected

corrected

Page 17 line 14: the sentence structure is awkward here.

corrected

Page 20: In the caption to figure 13, Indian is spell wrong. Also, why not show the global freshwater transport (black line) in panel b?

Indian corrected. After long discussions we have included the global line. No doubt, that the global line in general is helpful. Drawback is that the additional line reduces somewhat the readability of this figure.

**The climate of a retrograde rotating Earth**
**Response to Reviews**

Uwe Mikolajewicz and Co-authors

August 2018

**Reply to Reviewer 2**

The authors use a Earth system general circulation model allowing considerable process complexity (although at a lower spatial resolution than the current state of the art) to simulate the climate of a counter factual Earth that rotates around its axis in the opposite direction. The study is fascinating in conception, and appears to have been carried out and presented in a clear, careful and scientific manner. I have some comments and queries on specific matters of course, but the paper is sound and could really be published with only minor revision.

We thank the reviewer for the positive feedback and for the helpful and constructive comments. In the following we addressed all the points raised by the reviewer.

The biggest single improvement, in my opinion, would come from the authors being little clearer on the intended aims and conclusions from the study. Given our imperfect understanding of and ability to simulate the world as we see it now, it is obviously a bit of a stretch to apply precisely the same tools to a world for which we have no observational verification or ability to conclusively falsify hypotheses. I don't think that means that we shouldn't carry out an exercise like the one that has been presented here, but having a very clear framework in place, eg. "we want to test our general understanding of a certain feature of the real Earth", or "we're simply interested in exploring what such a place would look like for its own sake" - both would be quite valid, as far as I'm concerned - is helpful in interpreting what is shown. The current manuscript feels like it tries to do both at once without clearly framing the context for some of the results, and I'm left unsure, for instance, whether certain aspects of the write-up are really meaningful - one's interpretation depends on the context in which you read them. For instance, in the RETRO-S experiment, the Coriolis force is reversed but the direction of the diurnal solar cycle is left alone. The configuration is totally artificial and the physics in question is almost certainly not being reproduced "realistically" (even very high resolution NWP models have difficulty getting the physics involved in the daily timings of the afternoon rain in Africa right), so it's questionable that this has direct relevance to learning about our world. But it's an interesting, physically valid part of the artificial

construct we're being presented with when seen outside of a real-world context - I think the same sort of comment applies to the climate sensitivity section. The opposite is however true for some of the land-cover and ocean biogeochemistry results, where the RETRO model is tied to real-world distributions of $CO_2$, soil, ice and dust and it's difficult to see what results as being internally consistent in the context of a backwards-rotating ecosystem.A symptom of this may be that almost the entirety of the Conclusion section is a descriptive summary of what has been seen (and already described) in the simulation, rather than making any attempt to help the reader see what has been learned from the exercise in toto. Apart from having certain aspects of this fascinating Gedankenexperiment demonstrated for real, I'll admit to personally being unsure what robust conclusion might be taken from this (other than that playing with climate models is fun), so any help the authors could offer would be appreciated.

We agree with the reviewer, that the aims of the study were not as clear formulated as they could be. This goes back to the comprehensive nature of the study, where we tried to analyze all components of the climate system in order to get a full picture of the changes. Our first priority of the study is a more comprehensive understanding of the general features of the Earth, specifically in respect to spatial asymmetries and circulation regimes, as is highlighted in the introduction of the manuscript. To emphasize this, we changed the conclusions and moved some of the sensitivity experiments, which were conducted in order to get a deeper understanding into the different factors controlling the climate under a reversed rotation, to the Appendix. We also connected the individual parts better, in order to point out interactions between the components of the Earth system.

pg1, paragraph 2: some of the summarised effects and causes in the abstract don't come out very clearly in the main body of the manuscript itself, requiring the reader to piece together material in the different sections and appendices for themselves to make a picture that matches up with the abstract. Some of this could be explicitly outlined in the Conclusions section

We have attempted to address this by sharpening the presentation of the abstract and conclusions, but also, and more substantially, by moving some of the text and figures (on storm tracks and moisture transport) from the supplementary material into the main text.

pg2, line 29: the literature on monsoons is vast, I find it difficult to believe there is only one study looking at what sets their locations

Yes the monsoon literature is vast, but very little of it focuses on the general causes for the zonal distribution and extent of the monsoons and their relationship to the deserts. This contrast, between the large literature on the topic of monsoons and the relative sparseness on questions related to its zonal extent was actually what we had hoped to highlight as the simulations have a bearing on this. To make this clearer we have slightly rephrased this discussion.

p3,l27: Does "These simulations did not [...]" refer to just the simulations of Kamphuis et al, or of the results of both Smith and Kamphuis taken together?

Here we mean the two experiments of Kamphuis et al. We reformulated the sentence to: "... *This study did not, however, show evidence of a complete*

*reversal in the role of sub-polar North Atlantic and Pacific..."*

p3,l34: "for the most part" is very imprecise

Rephrased to: *"[...] and wherein most components in the Earth's oceans and biosphere (terrestrial land surface) were allowed to freely adapt to the changing conditions [...]"*.

p4,l13: it would be helpful to say where the atmosphere model top is, in terms of how much of the stratosphere is being modelled.

We added that the atmosphere reaches up to 10hPa in the text.

p4,l[20—26]: is the "long" model spinup (l20) different from the 6990 year model integration (l26), and if so, how long was it?

Yes, there was an additional spin up of almost 8000 years with some changes in the marine biogeochemistry model component during the run. This is now mentioned in the model and experiments section.

p4,l30: how one interprets the impact of not allowing ice, GHGs and aerosols to change very much depends on how you frame the results of the simulation - see above. The significant changes simulated in biological production and ocean deepwater would be expected to have an impact on the overall GHG concentration in the atmosphere, with a possible first order effect on the climate and possible major ice-sheet feedback - doing this sort of thing with enough detail is clearly out of scope for our present modelling capability, but it would be fun and informative to have some speculation from the authors!

Planetary-scale features of the marine biogeochemistry such as zonal and global means (e.g. net primary production, air-sea carbon fluxes) are largely unaffected by the direction of Earth's rotation. Hence, marine carbon inventory increases only by 100 PgC in RETRO compared to CNTRL. Given the ocean buffer capacity this would correspond to a decrease of atmospheric $CO_2$ concentrations of the order of 8 ppmV. However, in a system with a fully coupled carbon cycle any increase of the land carbon inventory with a corresponding impact on the atmospheric CO2 would be, to first order, compensated by the ocean. The projected change on land (increase by 86 PgC) is rather small compared to the marine carbon inventory ($\sim$ 38000 PgC). Thus, we expect only a rather small change of 1-2 ppm in a fully coupled simulation and consequentially very minor effects on climate. We deleted the sentences on the potential impact on the atm. $CO_2$ in section 4 and included a brief discussion in the conclusion section.

For the ice sheets we have looked into summer snow depth and temperature fields, from which we expect that prescribing present-day ice sheets does not seem to have a major impact on both simulations. We added a short paragraph at the end of section 4:

*Another prescribed present day feature are the ice sheets. In the RETRO and CNTRL simulations, no permanent snow cover was simulated outside of the prescribed glaciated areas (not shown). Additionally, the simulated summer temperatures over the ice sheets (not shown) do not indicate a strong inconsistency between the prescribed ice sheets and the simulated climate both for RETRO and CNTRL. Therefore, we did not find a major inconsistency from the prescribed present-day ice sheet configuration and the simulated climate*

p5,l3: piControl needs explaining

We added 'pre-industrial conditions' for clarification.

p5,l4: has ESM been defined yet?

Thanks for pointing this out. We now introduced ESM (Earth System Model) in the beginning of this section.

p6,l16: why is the Pacific cold tongue so much less evident in RETRO?

If you would expect a Pacific cold tongue in RETRO, for obvious reasons it would be in the West Pacific. The Indopacific acts here largely as one ocean, so the Indian Ocean would be the right place to search for a cold tongue. Indeed the Indian ocean has overtaken to a large extend the function of the tropical East Atlantic with strong upwelling.

Due to the presence of landmasses around the Northern Indian ocean, a strong monsoon signal occurs, which does not allow the cold tongue signal to develop.

p18,l29: could the authors speculate from the published model specifications and results why each model sees the MOC changes it does?

Each model is different and shows different responses, as can be seen in a variety of model inter comparisons. E.g. Cheng et al. (2013) showed that a weakening of the MOC varies considerably among different models under similar concentration pathways (RCP4.5, RCP8.5). They analyzed 10 different CMIP5 models and find that for RCP4.5 a weakening ranges between 5% and 40% between models and for RCP8.5 between 15% and 60%. Reasons for these differences can be attributed to differences in the gyre locations, deep water formation areas, precipitation and evaporation, etc. Such model dependent differences make it difficult to evaluate why we see exactly these differences.

This is in line with findings from other model inter-comparison studies investigating the response of the AMOC to perturbations, e.g. Gregory et al. 2005, or Swingedouw et al. 1013 or 2015). In all these studies it has not been possible to derive a clear explanation for the differences in the AMOC sensitivity of the different models, although the complete model output in some of these studies was available.

Here the situation is similar like in a warming climate, all models show changes into the same direction, but the magnitude of the response is different. A speculation on the reasons for the differences between different models would lack substance.

- Gregory, J. M., Dixon, K. W., Stouffer, R. J., Weaver, A. J., Driesschaert, E., Eby, M., Fichefet, T., Hasumi, H., Hu, A., Jungclaus, J. H., Kamenkovich, I. V., Levermann, A., Montoya, M., Murakami, S., Nawrath, S., Oka, A., Sokolov, A. P. and Thorpe, R. B. (2005) A model intercomparison of changes in the Atlantic thermohaline circulation in response to increasing atmospheric CO2 concentration. Geophysical Research Letters, 32 (12). doi: https://doi.org/10.1029/2005GL023209

- Swingedouw, D., C.B. Rodehacke, S.M. Olsen, M. Menary, Y. Gao, U. Mikolajewicz, and J. Mignot (2015) On the reduced sensitivity of the Atlantic overturning to Greenland ice sheet melting in projections: a multi-model assessment. Climate Dynamics 44, 3261-3279. doi:10.1007/s00382-014-2270-x.

- Swingedouw, D., C.B. Rodehacke, S.M. Olsen, M. Menary, Y. Gao, U. Mikolajewicz, and J. Mignot (2015) On the reduced sensitivity of the Atlantic overturning to Greenland ice sheet melting in projections: a multi-model assessment. Climate Dynamics 44, 3261-3279. doi:10.1007/s00382-014-2270-x.

- Cheng, W., Chiang, J.C.H., Zhang, D. (2013). Atlantic Meridional Overturning Circulation (AMOC) in CMIP5 Models: RCP and Historical Simulations, J.Clim, 26, 7187-7197, doi.org/10.1175/JCLI-D-12-00496.1

p21-24. Section 6 describes one of the most interesting aspect of the results and the importance of using an ESM, rather than a purely physical climate model, to do this kind of simulation. It's a real shame that the connections aren't there to either include the impact of changes in land cover and dust on the biogeochemistry, nor the follow on impact on ocean CO2 sequestration of the biogeochemistry and physical ocean changes together. The authors end by claiming that they expect that their model limitations don't affect the main features of the ocean carbon cycle, but I'm much less sure, and I'd welcome either more justification of this statement, or some interesting speculation on what could happen instead.

In agreement with the reviewer we certainly would have favored a consistent run with full carbon cycle coupling and a dust input field matching the projected pattern changes in land cover and atmospheric wind/precipitation fields. However, this is far beyond outlay of this study. As outlined in the comment above we speculate that the rather small changes in the carbon inventories of the land and marine realm would lead to only small changes in the atmospheric $CO_2$ concentration and, thus, would have no or a very minor effect on climate in a fully coupled set up. With respect to the impact of a fixed dust field on cyanobacteria growth in the Northern Indian Ocean we added a back of the envelop calculation to compare the iron supply from upwelling versus atmospheric input (Appendix D). We find that a large fraction (40-80%) of the iron demand of cyanobacteria could be delivered by ocean upward transport of the almost constant background iron concentration. We changed the main text correspondingly.

p27. Section 7 feels unfocused, and (given the main "framing" comment above) I'm really not sure what one is supposed to take from it

According to the reviewers suggestions we decided to move all perturbation studies to the appendix. Both chapters in Section 7 are sensitivity experiments, conducted to better understand the model simulations. The RETRO-S experiment allows us to estimate how much of the climate signal can be contributed to the rotation and how much to the reversal of the sun path. The section on climate sensitivity is important to estimate the response of a model to changing climate conditions. Given that the mean climates in RETRO and CNTRL

are very different we consider it interesting whether the climate sensitivity, and therefore feedbacks and interactions in the model system, change.

p28,l7: I don't think "numerous" sensitivity experiments have been shown, just 2: RETRO-S and a 4xCO2 run?

We agree with the reviewer and removed the word "numerous" here, as only two sets of sensitivity experiments were conducted.

p28,l12: I also don't think one can credibly call ice-sheets, soils and GHGs "minor" constituents of the Earth system!

We replaced "minor" with "other".

p30, Appendices: In my opinion some of the most interesting analyses have been relegated to relatively brief descriptions in the appendices. Personally, I would rather see all of this material in the main text in preference to the current section 7, although some of it may be a bit specialised and the Editor may want to take a view based on the intended audience for the journal. The other issue would be whether the features are really modelled robustly enough to be part of the main attraction - the functioning of an ENSO and its teleconnections on a RETRO planet could probably be a paper all on its own rather than 18 lines in an appendix, if the relevant physics in the model could be shown to be robust enough.

We tried to strike a balance in how much weight we gave each section so as to better emphasize the integrated aspects. But given some ambiguity in how best to do this, the reviewer's input was helpful and suggested that we could add more material describing the atmospheric circulation with out overloading the reader. Therefore, we moved key results from former appendices A (storm tracks) and B (moisture transport) to the main section of the manuscript. As the reviewer also correctly pointed out, a complete analysis of teleconnections and ENSO would be too comprehensive and out of the scope of the current manuscript. However, we think that including a short note, as is done in the Appendix, is still interesting to the reader who is interesting in this topic. Therefore, we would like to leave the former Appendix C as is.

p34,l8: How many years of data have been used in the power spectrum? The confidence intervals don't look very convincing - if the 1-10yr signal in CNTRL is supposed to look significant then so is 10+ in RETRO...?

For this analysis 1000 years were used. For CNTRL we see clearly more energy in the band 1-10 years, especially compared to periods longer than 20 years (about 1/2 order of magnitude). For RETRO we do not see more energy in a specific frequency band for the period range from 2 to 100 years.

The authors seem to have a habit of trying to make sentences out of phrase fragments that have no main clause eg pg3, line19 "One which includes different net freshwater forcing regimes for the ocean, while keeping the present continental geometry.", p6,l34 "The area with the largest amount of annual precipitation being near Ascension Island in the Southern Tropical Atlantic (...).". There are several occurrences of this construction throughout.

We have made changes to the specific sentences pointed out by the reviewer and several other places throughout the manuscript.

Several features are described as (objectively) "surprising" or "(un)expected", which are subjective assessments

We removed most occurrences of the word surprising. 'Expected', however, is left in the paper, as it indicates deviations from what we would expect from the classical theories.

pg2, line27: "that what would be caused" - what should be which

Thanks, we changed this.

p3,l5: "how the ocean transport heats"

Changed.

p3,l[11—15]: "stabeliz[es—ing]"

Changed.

p3,l25: "FAMOUS" needs an explanation/citation if you're going to specify the model

Added "with the climate model FAMOUS (Smith, 2012)" to clarify.

fig 1. why does the time axis go from 1000-8000 for runs that are described as 6990 years?

The original time axis referred to model years. For technical reasons we have to start in year 1010, so the real model years of the simulations are 1010 to 7999. We agree with the reviewer that this is confusing for the reader and have changed the time axis to start at 1.

p4,l7: "MPI-ESM" and "CMIP6" need explanation/citation

MPI-ESM and CMIP are now introduced and relevant citations added.

p4l13: "vertic"

Changed.

p5,l12: "mid-altitudes have surface easterlies

Changed.

p6,l2: "feint" should be "faint"

Changed.

p6,l16: "the American's warm"

Changed.

p6,l34: "like presentday Palau"

Changed.

fig3: "(10 Mm versus 8Mm)" I don't understand this part of the caption

The figure caption has been revised. The phrase in question hinted towards the different scale of northern and southern hemispheric plots. Probably more confusing than helpful, therefore we decided to remove it.

fig4: whilst very pretty, this figure lacks adequate labels as to what the flux components are, what time averaging has been applied, and takes up a lot of space illustrating something that "hardly differs from that of the CNTRL" and (pg9,l10) whose quantities might be expected to vary most interestingly on a hemispheric basis, rather than the global average actually shown

Yes, we agree it is a lot of real-estate to show very little change. But we wanted to draw attention to how similar the energy budget is despite large regional differences. This question has resonance when one considers that most of what we are sure about in terms of global warming is the global mean changes

to the energy budget, and hints at how different circulations can be for the same global mean.

p11,l3: new paragraph at "Changing the planetary rotation [...]"

Changed.

p12,l11: "vareity"

fig11,13: "Indic" Indian Ocean?

Thanks. Changed.

p21,l27: "[...] occur only very localized."

Changed into ' only few regions exist, where..'

p24,l10: Despite the fixed atmospheric CO2, the changes in the land biosphere in section 4 are extrapolated to an estimate of their potential impact on the CO2 concentration in the atmosphere - could a similar estimate not be made from the changes in the ocean carbon inventory?

We moved the discussion on potential implication of changes in the land and ocean carbon inventories to the conclusion section.

p29,l16: "Euro-Africa, with American" doesn't need the comma

Thanks, changed.

p29,l20: "Other studies" needs references

We included a reference to the Warren 1983 paper.

p33,l1: "Kongo" Changed into 'equatorial Africa', now in main text.

p33-35: figs C2,C4 not actually referred to in text. the former fig. C2 has been removed, the former fig. C4 was actually referenced, but with the wrong fig. number. This has been corrected.

p34,l4: "In RETRO" is repeated in this sentence

Corrected.

p37,l13: "n/a-n/a" in citation

Corrected.

p38,l[25—32]: two web links in citation

Corrected.

p39,l11: two web links in citation

Changed.

[revised manuscript text omitted]

**Appendix E: Storm track changes**

~~Given the change in the planet rotation and the inversion of prevailing winds in the upper atmosphere we expect the storm tracks to be modified. However we do not know, whether the change would be just zonally symmetric or instead involve a modification of storms intensity and lifetime. In order to asses that we perform a tracking using the Mean-Sea-Level Pressure (MSLP) data every 2 hours. Storms are defined as local minimum of MSLP and are tracked during their lifetime using the methodology described in Hoskins and Hodges (2002).~~

 In this section, additional results expanding on the tracking of cyclones in the northern hemisphere during boreal winter (DJF, section 3), as well as the analysis for the southern hemisphere during austral winter (JJA) are presented. Figure E1 shows the zonal averaged statistics of cyclones tracked in the northern hemisphere.

~~Track density (number of tracks per season for a unit area equivalent to a 5 degree spherical cup) computed for the northern hemisphere during DJF season. The data are seasonally averaged over the last 100 years of each simulation. Shown in the leftmost plot is the difference between RETRO and the CNTRL planet. As expected, cyclones are now forming over the eastern Atlantic and Pacific and are subsequently transported north-westward, instead of north-eastward as in CNTRL. It is interesting to note how both the Mediterranean and north America are strongly affected by the increase in track density.~~

[Figure]

**Figure E1.** Zonal averaged statistics obtained from the tracking algorithm for the DJF season in the northern hemisphere.

Results show that, while the storm tracks shift from the ocean to the land (Fig. 4), there is little difference in the zonal mean
5    statistics between RETRO and CNTRL besides of the baroclinic growth rate.

In the southern hemisphere, cyclones  shift from the Indian and Atlantic ocean to the Pacific ocean, as presented in Fig. E2. Consistent with that, the cyclones in the Pacific Ocean in the retrograde planet are stronger, have higher growth rates,

[Figure]

**Figure E2.**  As in Fig. 4 but  the southern hemisphere during the JJA season.

 and are also faster ( Fig. E3). However, these local changes are compensated such that the overall zonally-averaged statistics in the southern hemisphere do not change significantly (not shown).

10    **Appendix F:**

(Keith, 1995)
15

[Figure]

**Figure E3.** Zonal averaged statistics obtained from the tracking algorithm for the JJA season in the southern hemisphere Pacific Ocean region.

~~The reversal of the westerlies and trade winds in RETRO leads to a significant shift in the moisture transport. Due to the westerly trade-winds in RETRO, large amounts of moist-static energy are transported from the Atlantic towards North Africa and the Middle East. This transport persists through all seasons, except winter, when the trade winds are deflected southward across Nigeria towards Kongo and Tanzania (not shown). Further, moisture is transported from the Indian Ocean across India, Pakistan and the Middle East towards the Mediterranean Sea during all seasons except fall. The enhanced moisture flux towards North Africa and the Middle East from the Atlantic and Indian Ocean explains the significant greening in these areas (section 4) as well as the large net freshwater loss of the tropical Indian Ocean (
[revised manuscript text omitted]

---

## Author Response (AR2)

Dear Dr. Kirk-Davidoff,

we are pleased about the acceptance of the paper and submit a final version of the manuscript. We followed your suggestion and modified the sentence in question. Besides some changes in the acknowledgements the rest of the text is unchanged to the earlier version.

With best regards,
Uwe Mikolajewicz